# Quantifying degradation of the Imja Lake moraine dam with fused InSAR and SAR feature tracking time series

George Brencher[1], Scott T. Henderson[2,3], David E. Shean[1]

[1]University of Washington Department of Civil and Environmental Engineering, 1400 NE Campus Parkway, Seattle, WA 98195, USA
[2]University of Washington Department of Earth and Space Sciences, 1707 NE Grant Lane, Seattle, WA 98195, USA
[3]University of Washington eScience Institute, 1410 NE Campus Parkway, Seattle, WA 98195, USA

*Correspondence to*: George Brencher (gbrench@uw.edu)

**Abstract.** Glacial lake outburst flood (GLOF) hazards are often tied to the structural properties of the moraines that dam glacial lakes. Traditional investigations of moraine dam structure and degradation involve costly and logistically challenging in situ geophysical and repeat topographic surveys, which can only be performed for a small number of sites. We developed a scalable satellite remote sensing approach using interferometric synthetic aperture radar (InSAR), InSAR coherence, and SAR feature tracking to precisely measure moraine dam surface displacement and map the extent of buried ice. We combined time series from ascending and descending Sentinel-1 orbits to investigate vertical and horizontal surface displacement for the period 2017–2024 with ~12-day temporal sampling.

We applied our approach to quantify degradation of the Imja Lake moraine dam in the Everest Region of Nepal. We validated our SAR-based displacement measurements using 3D displacement measurements from very-high-resolution satellite stereo digital elevation models that span the study period. We found that a 0.3 km$^2$ area of the moraine dam has cumulatively subsided ~90 cm over the 7-year study period. Seasonal change in InSAR coherence provides evidence for buried ice throughout the moraine dam. We observed consistent downward and eastward displacement throughout the colder months, which we attribute to ice flow. The magnitude of downward vertical surface velocity increases in the warmer months, likely due to melting of buried ice. Our observations provide new insights into the timing and magnitude of the processes that control moraine dam evolution, with broader implications for regional GLOF hazard assessment and mitigation.

## 1 Introduction

Glaciers in High Mountain Asia (HMA), Earth's largest glacierized region outside the poles, are expected to lose between ~29% and 67% of their total mass by the end of the century (Hock et al., 2019; Rounce et al., 2020). Worldwide, glacier thinning and retreat are associated with an increase in the formation of glacial lakes (Shugar et al., 2020). These lakes constitute a significant hazard, as they can drain in sudden, catastrophic glacial lake outburst flood (GLOF) events that flood downstream valleys and can result in loss of life and damage to infrastructure (Mool et al., 2011; Riaz et al., 2014; Carrick and Tweed,

2016). For example, the September 2023 GLOF event in Sikkim, India, destroyed the Teesta III hydroelectric dam, washed away 15 bridges, stranded 3,000 tourists, and left at least 74 dead with more missing (Ali Badal, 2023; Choudhury and Hussain, 2023; Kumar and Travelli, 2023; Sebastian, 2023).

Effective management of GLOF hazards involves 1) identification and monitoring of glacial lakes that pose a significant hazard, and 2) mitigation of this hazard before GLOFs occur, potentially through engineering solutions. The former has been primarily accomplished through hazard assessments ranging from a single lake (e.g., Rana et al., 2000; Budhathoki et al., 2010; Wang et al., 2018; Sattar et al., 2021) to regional scale (e.g., Mool et al., 2011; Wang et al., 2012; Fujita et al., 2013; Allen et

al., 2016; Rounce et al., 2017). Glacial lakes can be dammed by moraines, bedrock, and glacier ice (Rick et al., 2022). Where glacial lakes are dammed by moraines, hazard assessments frequently consider moraine dam stability, the presence of buried ice within moraine dams, potential GLOF triggering events, and downstream impacts (Rounce et al., 2016). Where moraine dam instability is not identified as a primary GLOF trigger mechanism, melting of buried ice can increase lake area, reduce dam width and height, and provide potential pathways for seepage and piping (Richardson an Reynolds, 2000; Emmer and

Cochachin, 2013; Medeu et al., 2022). As such, the evolution of moraine dams may substantially increase GLOF likelihood.

As glacial lakes, moraine dams, and the surrounding landscape change over time (Huggel et al., 2010; Shugar and Clague, 2011; Kellerer-Pirklbauer et al., 2012; Ravanel et al., 2018), hazard assessment and mitigation tasks require repeated observations (Fujita et al., 2009; Rounce et al., 2017; Emmer et al., 2018). Satellite remote sensing has been used to create

glacial lake inventories, track glacial lake development (e.g., Fujita et al., 2009; Nie et al., 2018; Shugar et al., 2020), and recently, to monitor glacial lake dam and bank evolution (Haritashya et al., 2018; Scapozza et al., 2019; Wangchuk et al., 2022; Yang et al., 2022; Jiang et al., 2023; Yang et al., 2023; Yu et al., 2024). These remote measurements of surface displacement can be used to assess moraine dam stability, infer moraine dam structure, and quantify changes in moraine dam topography, providing critical information for GLOF hazard assessments and potentially guiding prioritization for in situ

surveys and hazard mitigation strategies.

## 1.2 Remote sensing of moraine dam degradation

C-band SAR can operate during both day and night, penetrate clouds, and penetrate thin, dry snow (Bürgmann et al., 2000; Sun et al., 2015). Both InSAR and SAR feature tracking (in original range–azimuth coordinates) can provide measurements of surface motion in the radar line-of-sight (LOS) direction.

InSAR relates the phase offset from successive SAR acquisitions to surface displacement, potentially with mm-level accuracy (Bürgmann et al., 2000; Rosen et al., 2000). InSAR has frequently been applied to measure displacement of ice-rich features, including glaciers and ice sheets (Massonnet and Feigl, 1998; Rosen et al., 2000), rock glaciers (Bertone et al., 2022), and

permafrost (Zhang et al., 2022). Recently, InSAR has also been applied to quantify surface movement of glacial lake dams (Scapozza et al., 2019; Yang et al., 2022; Jiang et al., 2023; Yang et al., 2023; Yu et al., 2024).

While InSAR offers unparalleled precision, several factors can reduce measurement accuracy, including atmospheric noise, layover, and radar shadow. InSAR requires 1) coherent surface change with similar scatterer characteristics between radar acquisitions, and 2) sufficiently small phase-change gradients between adjacent pixels to allow reliable phase unwrapping (Itoh, 1982; Handwerger et al., 2015). While errors caused by atmospheric conditions and acquisition geometry can be avoided or corrected, errors caused by rapid surface change may cause underestimation of true LOS surface velocity. InSAR coherence, a unitless measure of the "sameness" of surface scatterers between acquisitions, can be used to discriminate between reliable and unreliable InSAR measurements (e.g., Schmidt and Bürgmann, 2003). InSAR coherence can also be used to identify significant change in surface characteristics, and low coherence has been used to map the extent of desert erosion (e.g., Cabré et al., 2020), landslides (e.g., Ohki et al., 2020; Jacquemart and Tiampo, 2021), flooding (e.g., Chini et al., 2019), and debris-covered glaciers (e.g., Atwood et al., 2010; Frey et al., 2012; Lippl et al., 2018).

Feature tracking (pixel offset) techniques offer an alternative method to retrieve surface displacement measurements from SAR data when InSAR is not possible due to loss of coherence or large displacement gradients between adjacent pixels. Feature tracking involves 2-D cross-correlation of backscatter amplitude values in successive radar images to measure offsets in the range (across-track) and azimuth (along-track) directions. This technique has been used to measure glacier surface velocity (e.g., Fahnestock et al., 1993; Strozzi et al., 2002, 2008; Fahnestock et al., 2016) and fast-moving features surrounding glacial lakes (Scapozza et al., 2019; Yang et al., 2022). While feature-tracking measurements offer lower resolution and precision than InSAR measurements, and measurement quality can be degraded by large surface changes, they are more robust to large surface displacements and rapid surface change (Joughin, 2002; Zheng et al., 2023).

Here, we present a method to create a single cumulative displacement time series from InSAR and feature tracking measurements using a variation of the small baseline subset approach (SBAS; Berardino et al., 2002; Schmidt and Bürgmann, 2003). The SBAS approach uses a network of displacement observations to solve for the displacement of a given pixel between successive acquisitions. It was developed for InSAR interferograms (Berardino et al., 2002) and has subsequently been applied to SAR feature tracking (e.g., Casu et al., 2011; Euillades et al., 2016; Guo et al., 2020; Samsonov et al., 2021) and optical feature tracking offsets (e.g., Bontemps et al., 2018; Altena et al., 2019; Lacroix et al., 2019). The SBAS approach, however, has issues during periods with poor coherence between successive acquisitions, which can introduce errors in the resulting displacement time series. This limitation has previously been addressed by constraining the time series inversion using geophysical constraints parameterized in time (López-Quiroz et al., 2009) and spatial covariance (Jolivet and Simons, 2018). We address this issue by constraining the per-pixel InSAR time series inversion with additional observations from SAR feature

tracking, which is more robust to decorrelation than InSAR. Our approach does not require prior assumptions about the displacement patterns of pixels over time.

Previous studies combined InSAR and feature tracking to produce self-consistent surface displacement maps over large ice sheets and ice caps, where surface velocity is often too fast for InSAR alone (e.g., Joughin, 2002; Gray et al., 2007; Liu et al., 2007; Sánchez-Gámez and Navarro, 2017; Joughin et al., 2018). To our knowledge, no studies have combined these techniques to measure year-round surface displacement of ice-cored moraine features, where we expect seasonal variability in surface melt rates and associated debris reworking.

**2 Imja Lake Study Site**

Imja Lake is a ~2.8 by 0.6 km glacial lake covering ~1.6 km$^2$ in the Everest region of Nepal (Fig. 1). Imja Lake is impounded on its north and south sides by lateral moraines, above which rise the steep slopes of adjacent mountain peaks: Imja Tse to the north and Ombigaichen to the south. To the east, the ~860 m wide calving front of the Lhotse Shar and Imja Glaciers terminates in the lake. Imja Lake formed in the early 1970s as supraglacial lakes coalesced on the stagnant, debris-covered tongue of Imja

Glacier. It expanded at a roughly linear rate of 0.02 km$^2$/yr until 1997, then at a rate of 0.03 km$^2$/yr from 1997 to 2020 (Watanabe et al., 2009; Gupta et al., 2023).

To the west, Imja Lake is impounded by an ice-cored terminal moraine covering 0.62 km$^2$. The Imja Lake moraine dam (hereafter, the moraine dam) has a low-relief, hummocky surface with ridges and furrows, ponds, and ice cliffs, with a mean

elevation of 4,982 m (orthometric height above the EGM2008 geoid). The lake drains across the moraine via a series of linked ponds. An artificial drainage channel was constructed downstream of the westmost pond in 2016 (UNDP, 2012; Khadra, 2016). The moraine dam has an enclosed concave-up surface and broadly slopes down toward the lake and surface ponds (Fig. S1). Immediately northwest of the moraine dam is a lobe of the debris-covered Lhotse Glacier.

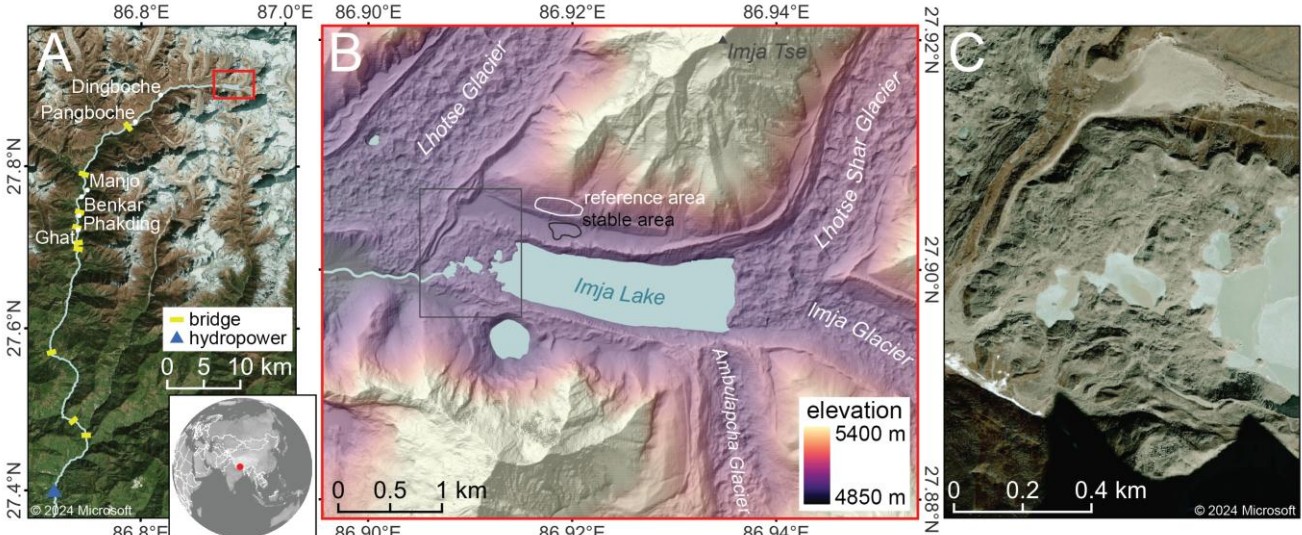

**Figure 1: Context maps for the Imja Lake moraine dam. A) Microsoft satellite basemap showing the area downstream of Imja Lake, including settlements and infrastructure along the Dudh Koshi river. B) Color-shaded relief map from median composite of EarthDEM strips acquired during the 2014-2019 period (Porter et al., 2022). C) © Microsoft satellite basemap detail of the Imja Lake moraine dam. Microsoft product screen shots reprinted with permission from Microsoft Corporation.**

Geophysical studies using ground-penetrating radar (GPR) and electrical resistivity tomography (ERT) have documented the

125 extent and thickness of buried ice in the moraine dam, with continuous buried ice tens of meters thick to the east transitioning

to discontinuous blocks of buried ice to the west (Hambrey et al., 2008; Somos-Valenzuela et al., 2012; Dahal et al., 2018).

Two studies quantified Imja Lake moraine dam degradation rates using in situ measurements obtained with traditional and

Global Positioning System (GPS) surveying techniques (Watanabe et al., 1995; Fujita et al., 2009), and one study measured

moraine dam displacement by differencing the 30-m Shuttle Radar Topographic Mission Global 1 (SRTM-GL1) DEM

acquired in February 2000 with a 2-m WorldView stereo DEM acquired in February 2016 (Haritashya et al., 2018). These

studies document meter- to sub-meter-scale subsidence and horizontal motion over the surface of the moraine dam, with the

fastest motion occurring in the northeastern area.

## 3 Data

We downloaded all available Copernicus Sentinel-1 C-band single-look complex (SLC) images collected over the study site

from January 1, 2017 to March 1, 2024. The revisit time for this period was generally 12 days; we did not include data before

2017 due to longer gaps between acquisitions. All images were acquired in interferometric wide (IW) swath mode with vertical

co-polarization (VV) along ascending (satellite moving north and looking east) relative orbit 12 and descending (satellite

moving south and looking west) relative orbit 121. To minimize download time and storage, we limited processing to SLC

bursts covering our study site, with a total of 214 ascending bursts (Burst ID 012_023790_IW1) and 227 descending bursts

(Burst ID 121_258661_IW2). Burst dimensions are ~20 km in azimuth (along-track direction) by ~85 km in range (across-track direction), with pixel spacing of 14.1 m in azimuth and 2.3 m in range (Fig. S2).

We used the 2022_1 release of the 30-m Copernicus GLO-30 DEM (European Space Agency, 2021) to remove the topographic component of phase and geocode the SAR data products. The GLO-30 DEM product has an absolute vertical accuracy (LE90)
of <4 m and an absolute horizontal accuracy (CE90) of <6 m (NSSDA, 1998; European Space Agency, 2022).

We downloaded daily 2-m air temperature and total precipitation data from the Pyramid Weather Station (UCAR/NCAR-Earth Observing Laboratory et al., 2011), which is located ~11 km northwest of the Imja Lake moraine dam at an elevation of 4951 m, about 30 m lower than the elevation of the moraine dam. We aggregated available daily measurements between October 1,
2002 and December 31, 2009 to derive monthly climatology (mean, minimum, maximum and standard deviation) for the ~7-year period.

## 4 Methods

By combining InSAR and feature tracking measurements, we leverage the strengths of both methods to provide a more complete record and improved understanding of moraine dam kinematics. This approach captures large seasonal variations,
including slow cold-season displacement with InSAR and rapid warm-season surface change with feature tracking. However, feature tracking measures the displacement of discrete kernels of neighboring pixels, so it provides inherently lower spatial resolution than per-pixel InSAR displacements (Joughin, 2002). By combining InSAR and feature tracking measurements, we effectively degrade the spatial resolution of the InSAR measurements in exchange for improved displacement accuracy. This compromise is essential for applications involving debris-covered ice, as large surface change during the warmer months
reduces InSAR coherence between Sentinel-1 acquisitions. It is also important to note that extreme, rapid and/or highly localized surface change from processes like iceberg calving, backwasting, and fast-moving debris flows will still present challenges for the combined approach, potentially resulting in underestimation of surface deformation. Nonetheless, combining InSAR and feature tracking offers improved displacement time series that are more accurate than those produced independently using either method alone.

### 4.1 SAR data processing

To extract surface displacement information from the Sentinel-1 SLC products, we generated both interferograms and feature tracking offsets using the Hybrid Pluggable Processing Pipeline (HyP3) ISCE2 Plugin (Hogenson et al., 2020), which enables batch processing with the Jet Propulsion Laboratory InSAR Scientific Computing Environment, ISCE version 2.6.3 (Rosen et al., 2012; Fig. 2).

We used the insar_tops_burst workflow to process interferograms with five looks (effectively, a spatial averaging factor of five) in the range direction and one look in the azimuth direction, geocoded to 20 m by 20 m pixels (Fig. S2; Fig. S3). To form redundant networks of ascending and descending interferograms, we created interferograms for each burst acquisition using the three nearest burst acquisitions in time. In total, we processed 636 ascending burst interferograms and 675 descending burst interferograms (Table 1).

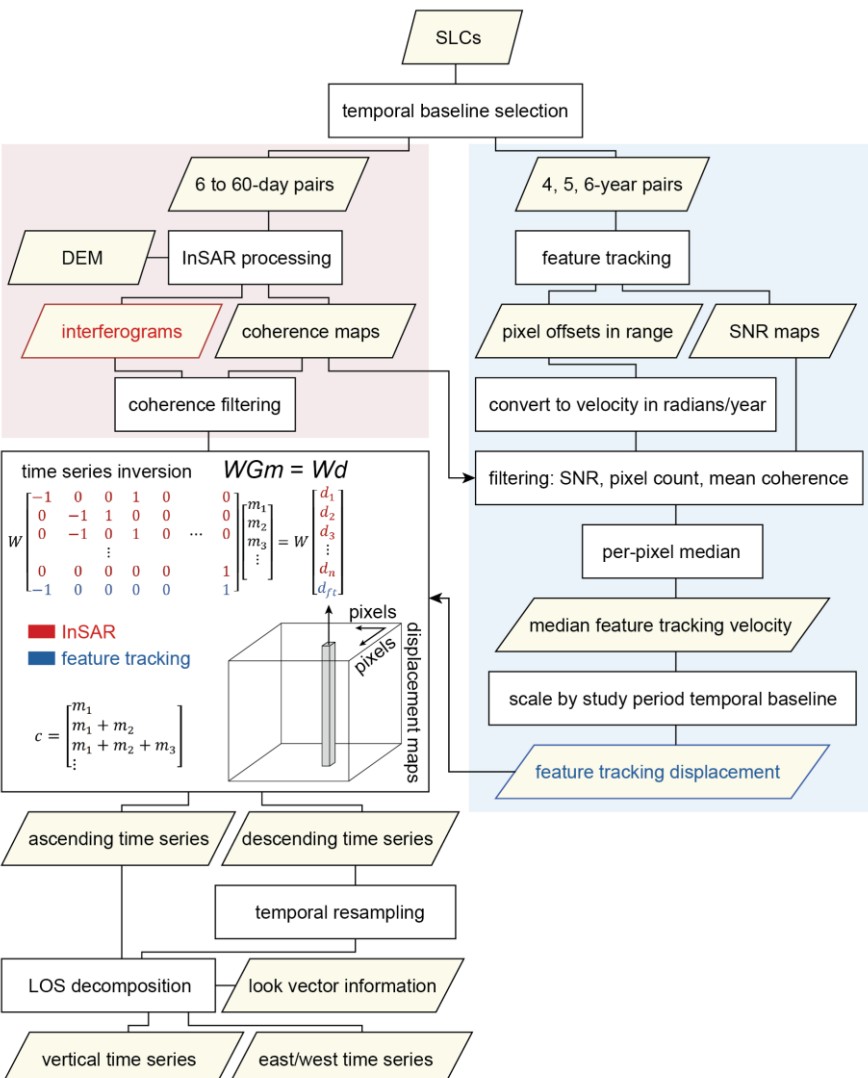

**Figure 2: Flowchart illustrating our combined InSAR and feature-tracking approach. The time series inversion schematic is adapted from Schmidt and Bürgmann (2003). The equation relates the observed phase change ($d$) from each of $n$ interferograms and a single cumulative feature tracking displacement product ($d_{ft}$) to the change in phase between acquisitions ($m$) at a given pixel. The design matrix $G$ describes the temporal baseline of each pair, with -1 corresponding to the primary acquisition date and 1 corresponding to the secondary acquisition date. The diagonal weight matrix $W$ includes coherence values. The cumulative phase change time series ($c$) is prepared from the inversion results for each pixel.**

We selected a local "reference area" and "stable area" (Fig. 1) as close to the moraine dam as possible, with high mean InSAR coherence for the entire study period. The reference area (~0.06 km$^2$, ~144 pixels) was used to remove atmospheric noise, while the stable area (~0.03 km$^2$, ~87 pixels) was used to assess derived measurement accuracy and precision.

We co-registered all SLCs and performed feature tracking on the SLC products in radar coordinates at the native resolution

using a modified version of the insar_tops_burst workflow and the 'dense ampcor' ISCE2 feature tracking routine (Fig. 2). We selected kernel sizes and skip sizes based on the characteristics of the moraine dam and an empirical sensitivity analysis. The moraine dam is roughly square (~800 by 900 m) in ground range. To maximize the number of unique feature-tracking measurements over the moraine dam that do not extend beyond its edges, we chose to use kernel sizes and skip sizes that were roughly square in ground range. Kernels that span the moraine dam and the surrounding area are likely to produce poor matches

due to intra-kernel variability in the magnitude and direction of surface displacement.

Smaller kernel sizes will result in poor matching, while larger kernel sizes will reduce the spatial resolution of the output feature tracking measurements. We found that a kernel size of 10 pixels in azimuth and 60 pixels in range (141 by 138 m in ground range) was the smallest possible square kernel size that provided high-quality matches. We used a skip size of 4 pixels

in azimuth and 20 pixels in range, or 56 by 46 m in ground range, to minimize interpolation without increasing processing time. Smaller skip sizes substantially increased processing time with negligible improvement, while larger skip sizes required more interpolation between sparse measurements. The search window size was set to 10 pixels in azimuth and 30 pixels in range, or 141 m by 69 m, which is much larger than the expected cumulative surface displacement of the moraine dam over the study period.

To estimate the accuracy of our feature tracking measurements, we first calculated the ground-range resolution of the slant-range SLCs. Following Notti et al. (2010) we computed the ratio of slant-range resolution to ground-range resolution (R-index) for the SLCs, considering terrain slope, terrain aspect, and the radar LOS vectors. We used this index to calculate the SLC resolution in the ground-range direction (Fig. S4). Over the moraine dam, the median ground-range resolution was 4.38 m for

the ascending SLCs and 3.82 m for the descending SLCs. Assuming perfect SLC co-registration and a conservative feature-tracking precision of 0.2 pixels (Fialko et al., 2001; Strozzi et al., 2002), the theoretical accuracy of our feature tracking measurements is 2.82 m in azimuth, 0.88 m in range for the ascending SLCs, and 0.76 m in range for the descending SLCs. Given the expected mean surface velocity of <1 m per year for the moraine dam (Fujita et al., 2009), we generated feature tracking offsets with multi-year temporal baselines to maximize displacement between acquisitions. As longer temporal

baselines could result in degraded quality due to significant changes in surface scatterer characteristics, we generated offsets for all possible feature tracking image pairs with temporal baselines of 4, 5, and 6 years (Table 1). In total, we generated 192 ascending and 210 descending feature tracking offset maps with corresponding signal-to-noise ratio (SNR) maps. The expected

precision in the range direction for the 4-, 5-, and 6-year feature tracking offset maps is 0.22 m/yr, 0.18 m/yr, and 0.15 m/yr, respectively.

**Table 1. Summary of InSAR and feature tracking datasets.**

| Orbit | Acquisitions | Dataset | Pair count | Mean temporal baseline | Temporal baseline range |
|---|---|---|---|---|---|
| ascending track 12, Burst ID 012_023790_IW1 | 214 | InSAR | 636 | 24.3 days | 12-48 days |
| | | feature tracking | 192 | 4.7 years | 4-6 years |
| descending track 121, Burst ID 121_258661_IW2 | 227 | InSAR | 675 | 23.0 days | 6-60 days |
| | | feature tracking | 210 | 4.7 years | 4-6 years |

## 4.2 Time series inversion

Since both InSAR interferograms and the component of feature-tracking motion in the slant-range direction are measurements of displacement along the radar LOS (Fig. S5), these measurements can be combined to produce more accurate estimates of LOS surface displacement over time. We used the Miami INsar Time-series software in PYthon (MintPy) package (Yunjun et al., 2019) to prepare cumulative LOS displacement time series from all InSAR and feature tracking observations for each relative orbit.

To avoid cumulative displacement errors caused by poor feature tracking matches, we aggregated the available feature tracking offset products over time and prepared a single median velocity product for each relative orbit. These median velocity products were then used to constrain the cumulative displacement solution from the network of InSAR interferograms from the same relative orbit (Fig. 2).

Several preprocessing steps were required to produce the median feature tracking velocity products for each relative orbit. We first converted all feature tracking offsets in range from pixels to C-band phase in radians, the same unit as our interferograms. We then masked all pixels with signal-to-noise ratio lower than 8 to remove unreliable measurements (Casu et al., 2011). This threshold was set to minimize the resulting magnitude of noise in the stable area. Next, we calculated the median surface displacement in range observed in the reference area and subtracted it from each offset map to mitigate potential co-registration bias. At this point we divided each feature tracking offset map in range by its temporal baseline and computed a per-pixel median rate for the entire time series stack, resulting in a single median velocity product for each relative orbit. We masked any pixels with less than 10 feature-tracking observations or a mean InSAR coherence greater than 0.85 during late summer (day of year 220–280, the lowest coherence period), as the InSAR measurements alone should provide accurate displacement time series for those pixels. We also computed the per-pixel standard deviation of the feature tracking offset maps to assess variability due to real changes and measurement error.

Prior to time series inversion, we removed interferograms with mean coherence less than 0.6 over the moraine dam, unless they were required to form a continuous network. This filter effectively removed interferograms impacted by intermittent snow accumulation, larger (>200 m) perpendicular baselines, or large surface change over the temporal baseline. To convert our per-pixel median feature tracking velocity (displacement rate) products into a cumulative displacement product analogous to an interferogram for the full 2017 to 2024 study period, we multiplied by the appropriate temporal baseline for the ascending (7.29 years) and descending (7.36 years) time series. The cumulative feature tracking displacement product was assigned a uniform coherence of 0.6, which is the minimum allowable mean coherence for the moraine dam area. The network of per-pixel displacement measurements, including the interferograms and the ~7-year cumulative feature tracking displacement product, was then inverted using phase variance, a measure of expected phase noise calculated using coherence (Tough et al., 1995; Hanssen et al., 2001), as a weight (Yunjun et al., 2019). As coherence is used to calculate phase variance, the cumulative feature tracking displacement product is weighted lower than interferograms where coherence is greater than 0.6.

Following the time series inversion, we computed and removed the median apparent displacement over the reference area from the LOS displacement estimate at each time step to mitigate atmospheric noise. The standard deviation of the median apparent displacement values over the reference area was 3.1 cm for the ascending data (n=214) and 0.5 cm for the descending data (n=227). The moraine dam is approximately 900 by 800 m and the reference area is only 250 m northwest of the moraine dam. Given the expected correlation length scales of several km for atmospheric noise in mountains (e.g., Bekaert et al., 2015), the apparent displacement in the reference area should be similar to the atmospheric noise over the moraine dam. To evaluate remaining measurement uncertainty, we first computed the mean apparent displacement in the stable area at each time step, then computed the standard deviation of this spatial mean over all time steps to characterize the magnitude of remaining atmospheric noise. The standard deviation was 0.2 cm for the ascending orbit and 0.4 cm for the descending orbit. Finally, we delineated a "moving area" polygon over the moraine dam including pixels with mean velocity of ~1 cm/yr above the background noise level in the ascending and descending cumulative displacement time series.

## 4.3 Surface displacement from LOS decomposition

Given two cumulative displacement measurements over the moraine dam, each with a different LOS from different relative orbits, we can solve for two components of the 3D surface displacement vector (Wright et al., 2004; Fuhrmann and Garthwaite, 2019). Since the mean LOS incidence angle at the site is 35.8° for the ascending burst and 37.8° for the descending burst, and the mean LOS azimuth (defined clockwise from north) is 79.3° for the ascending burst and 280.5° for the descending burst, we chose to solve for the vertical (up/down) and horizontal (east/west) surface displacement components. We assume that any north/south displacement would be negligible when projected into the available LOS vectors.

The LOS unit vector ($\hat{l}$) of each pixel can be defined as:

$$\hat{l} = \cos(\gamma)\sin(\theta)\,\hat{n} + \sin(\gamma)\sin(\theta)\,\hat{e} - \cos(\theta)\,\hat{z}\,, \tag{1}$$

for ascending ($\hat{l}_{asc}$) and descending ($\hat{l}_{des}$) geometry, where $\gamma$ is the LOS azimuth angle, $\theta$ is the LOS incidence angle defined relative to surface-normal direction, and $\hat{n}$, $\hat{e}$ and $\hat{z}$ represent the north, east, and up axes, respectively, which define the up/down ($\hat{l}_{ud} = 0\hat{n} + 0\hat{e} + 1\hat{z}$) and east/west ($\hat{l}_{ew} = 0\hat{n} + 1\hat{e} + 0\hat{z}$) unit vectors.

If we assume that the observed cumulative ascending ($d_{asc}(t)$) and descending ($d_{des}(t)$) displacement at time $t$ is only sensitive to the up/down ($d_{ud}(t)$) and east/west ($d_{ew}(t)$) surface displacement components, we obtain the following equations:

$$d_{asc(t)} = \hat{l}_{asc} \cdot \hat{l}_{ud} * d_{ud}(t) + \hat{l}_{asc} \cdot \hat{l}_{ew} * d_{ew}(t)\,, \tag{2}$$

$$d_{des(t)} = \hat{l}_{des} \cdot \hat{l}_{ud} * d_{ud}(t) + \hat{l}_{des} \cdot \hat{l}_{ew} * d_{ew}(t)\,, \tag{3}$$

We can then rearrange these equations to solve for the up/down and east/west displacement components:

$$d_{ud}(t) = \frac{\hat{l}_{asc} \cdot \hat{l}_{ew} * d_{des}(t) - \hat{l}_{des} \cdot \hat{l}_{ew} * d_{asc}(t)}{\hat{l}_{asc} \cdot \hat{l}_{ew} * \hat{l}_{des} \cdot \hat{l}_{ud} - \hat{l}_{des} \cdot \hat{l}_{ew} * \hat{l}_{asc} \cdot \hat{l}_{ud}}\,, \tag{4}$$

$$d_{ew}(t) = \frac{\hat{l}_{asc} \cdot \hat{l}_{ud} * d_{ud}(t) - d_{asc}(t)}{\hat{l}_{asc} \cdot \hat{l}_{ew}}\,, \tag{5}$$

To accomplish this, we interpolated the descending orbit cumulative displacement time series to match the acquisition dates of the ascending orbit time series, and then solved for the vertical and east/west displacement at every pixel at each ascending acquisition time.

To compute the vertical and east/west surface velocity, we divided the observed cumulative displacement by the time between acquisition dates. We aggregated these velocity component time series to compute the per-pixel monthly median values over the entire study period.

To identify and map areas of buried ice, we calculated a winter coherence map using the per-pixel median coherence for all 12-day temporal baseline observations from winter to early spring (day of year 0–100), and a summer coherence map using the per-pixel median coherence for observations in late summer (day of year 220–280). We attribute large observed seasonal decreases in InSAR coherence to surface changes caused by melting of buried ice during warm months.

### 4.4 Preparation of a high-resolution 3D displacement validation dataset from stereo DEMs

We identified and processed two Maxar WorldView-3 in-track stereo image pairs spanning the 2017–2024 study period, including the February 11, 2016 WorldView-3 stereo pair (catalog IDs 104001001854B000, 10400100175C2D00) used by Haritashya et al. (2018), and a new in-track stereo pair acquired on January 30, 2025 (10400100A0D45D00,

10400100A193EE00). We used the latest version (3.5.0-alpha) of the Ames Stereo Pipeline (Shean et al., 2016; Beyer et al., 2018) and the processing settings outlined by Bhushan and Shean (2021) and Bhushan et al. (2024) to prepare the corresponding DEMs with 1-m posting.

We co-registered the January 30, 2025 DEM to the February 11, 2016 DEM using the demcoreg (Shean et al., 2023) implementation of the Nuth and Kääb (2011) algorithm (Fig. S6). We then corrected residual in-track "jitter" and cross-track "CCD array geometry" artifacts (see Shean et al., 2016) by fitting a Savitzky-Golay filter (window of 101 px, 2nd-order polynomial basis function) to the median of elevation difference residuals over static surfaces for each row and column (Fig.

S7).

We prepared final cumulative elevation change and elevation change rate maps for the ~9-year period (Fig. 3C). The final elevation difference residuals over all exposed static control surfaces used during co-registration had a median of 0.00 m and normalized median absolute deviation (NMAD) of 0.14 m (<0.5 px) for the ~9-year period (Fig. S8), corresponding to a 1-

sigma uncertainty of ~1.6 cm/yr. The elevation difference residuals over the stable area used for calibration of the InSAR and feature-tracking results (see Fig. 1 for context) had a mean bias of -0.54 cm and a standard deviation of 7.77 cm, corresponding to apparent vertical displacement rates of -0.06 cm/yr and 0.87 cm/yr, respectively.

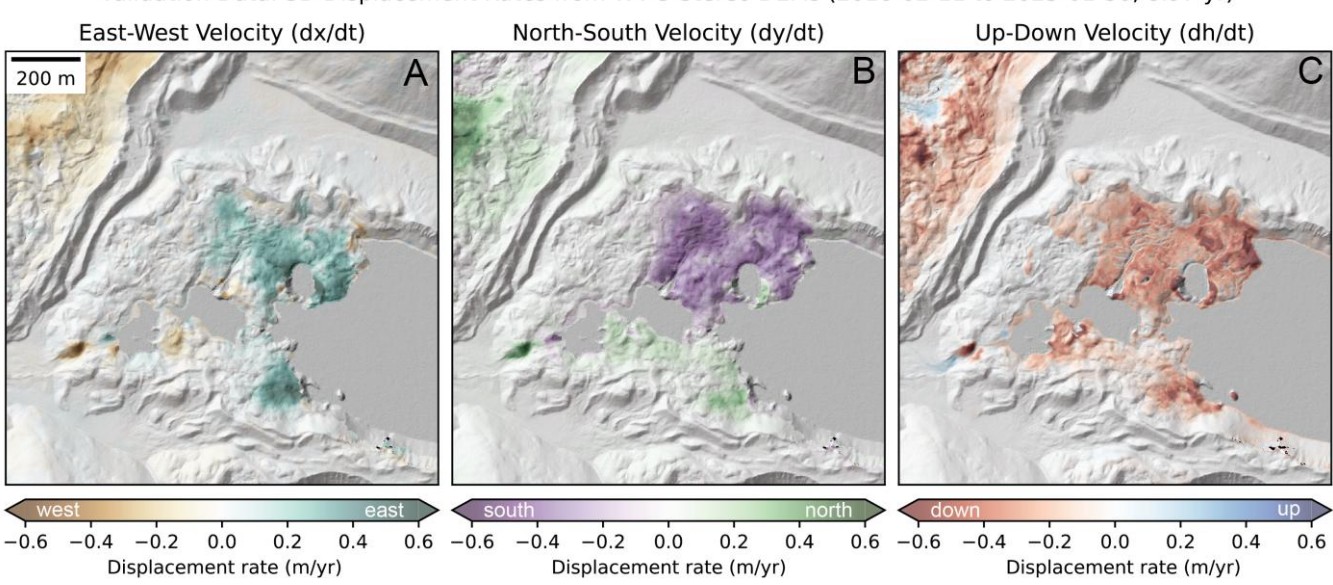

**Figure 3: Long-term (~9-year) east/west, north/south, and vertical surface velocity validation data, calculated from feature tracking**
**of co-registered 1-m DEM hillshade products and a vertical DEM difference product. These products document the spatial extent and magnitude of moraine dam surface motion during the study period with enough detail to capture signals associated with ice cliff retreat, downslope flow, and anthropogenic cut/fill and sediment deposition associated with the October 2016 lowering project (UNDP, 2012; Khadra, 2016). These products serve as validation for the combined InSAR and SAR feature-tracking time series, which offer detailed temporal evolution of these processes (Figures 4-5), with reduced spatial resolution.**

Following co-registration, we created shaded relief maps for the two DEMs using the "combined" hillshade option in the gdaldem utility (Rouault et al., 2024). We digitized the lake and pond shoreline using the January 30, 2025 orthoimages and shaded relief map, and masked the hillshade products over water. We then used the Ames Stereo Pipeline correlator to prepare dense horizontal sub-pixel displacement maps tracking surface motion between the hillshade products (disp_mgm_corr.py utility of Bhushan and Shean [2025]), following the methods outlined in Bhushan et al. (2024). The final horizontal velocity

(m/yr) components (Fig. 3A–B) were calculated based on the 1-m DEM pixel size and the ~9-year time period. To assess the uncertainty of the validation data, we computed the mean and standard deviation of the observed east/west (-1.5 cm/yr, 2.3 cm/yr, respectively) and north/south (-2.0 cm/yr, 1.9 cm/yr, respectively) velocity components over the stable area.

## 4.5 Validation of combined InSAR and SAR feature-tracking products

To enable direct comparison of our time series results with the DEM-derived validation data, we first masked pixels with a

340 cold-season coherence of <0.7 (Fig. 4). These areas contain rapid winter displacement caused by backwasting of ice cliffs, which have large signals in the DEM difference products that we do not expect to capture with Sentinel-1 SAR observations. We next resampled the 1-m DEM-derived products to match the 20-m grid of the Sentinel-1 SAR observations using median resampling, then smoothed the DEM-derived products to match the SAR feature tracking resolution using a 7 by 7 px (~140x140 m) Gaussian kernel. We computed zonal statistics (mean, standard deviation) for each velocity component

(up/down, east/west) over the moraine dam. We also computed $R^2$ values between the InSAR/SAR velocity component and the corresponding DEM-derived validation velocity component. Finally, we used the north/south component of DEM-derived velocity to assess expected bias caused by our up/down and east/west decomposition (See "LOS Decomposition bias" section of Supplementary Material for additional details).

## 4.6 Comparison with InSAR-only and SAR-feature-tracking-only SBAS time series

We prepared two additional time series from the same Sentinel-1 SLC products: a standard InSAR-only SBAS time series and a SAR-feature-tracking-only SBAS time series. The InSAR-only SBAS time series used the same processing parameters as described for our combined approach, but did not include the scaled median velocity product from feature tracking for the inversion (blue row in Fig. 2 system of equations). The SAR-feature-tracking-only SBAS time series included the same set of 4-, 5-, and 6-year temporal baseline pairs from our combined approach (192 ascending and 210 descending), as well as all

possible image pairs with a 3-year temporal baseline (182 ascending and 132 descending), to ensure that the system was overdetermined (total ascending $n=374$, descending $n=342$). As with the DEM-derived validation, we computed zonal statistics (mean, standard deviation) for each velocity component (up/down, east/west) for the InSAR-only and SAR-feature-tracking-only SBAS time series over the moraine dam.

## 5 Results

### 5.1 InSAR coherence

We examined the mean InSAR coherence from all pairs on both the ascending and descending orbits over the Imja Lake moraine dam during the full 2017–2024 study period (Fig. 4). The pair temporal baseline strongly controlled InSAR coherence over the moraine dam moving area (Fig. 4C, G), with shorter temporal baselines (i.e., 12 days) offering higher mean coherence than longer temporal baselines (i.e., 36–60 days). Over the stable area, we observed no obvious relationship between temporal baseline and coherence (Fig. 4D, H). Descending orbit coherence was on average slightly higher (mean 0.80) than ascending orbit coherence (0.74) over the moraine dam moving area for 12-day temporal baseline interferograms.

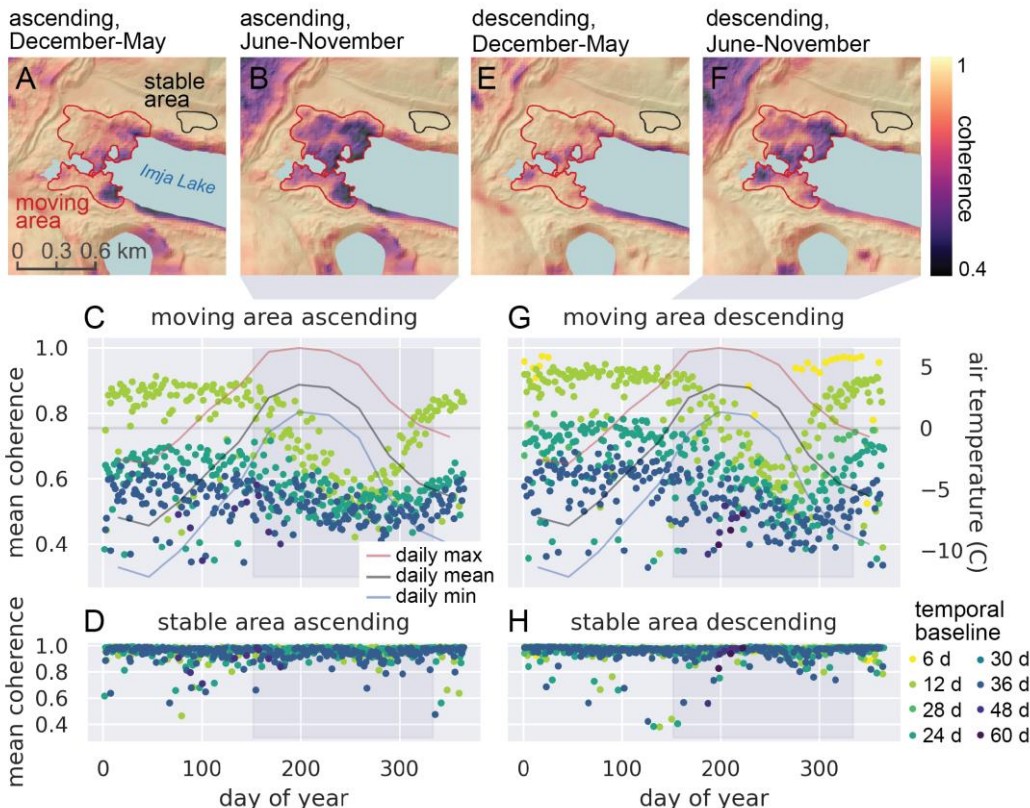

**Figure 4: Imja Lake moraine dam InSAR coherence and air temperature. Mean coherence maps for all 439 12-day pairs of ascending Sentinel-1 SAR images during the colder months (December-May, A) and warmer months (June-November, B). C) Mean coherence of the moraine dam moving area in all ascending interferograms. Lines show mean daily maximum, mean, and minimum air temperature for each month from the nearby Pyramid Weather Station (UCAR/NCAR-Earth Observing Laboratory et al., 2011). The gray line corresponds to 0° C. D) Mean coherence of the stable area for all ascending interferograms. Panels E-H show corresponding plots for descending pairs.**

We observed a systematic seasonal cycle in InSAR coherence values over the moraine dam moving area. Coherence remained largely stable from the beginning of the year until early June. For 12-day temporal baseline interferograms, mean coherence decreased by approximately 0.3 from June through mid-September. Coherence then increased from mid-September through

mid-November. This seasonal decrease in coherence lags about 60 days behind the timing of positive mean daily maximum temperatures at the nearby Pyramid Station (Fig. 4C, G). No significant seasonal coherence change was observed over the stable area (Fig. 4D, H).

The decrease in coherence is spatially constrained. Visual inspection of mean coherence maps from the annual high-coherence period (December–May) and low-coherence period (June–November) revealed that coherence decreased over almost all of the moraine dam moving area (Fig 4A-D). Several areas of low coherence in December–May expanded and displayed a further decrease in coherence during June–November. Areas of high-coherence in December–May also displayed a decrease in coherence during June–November. Low-coherence areas were mostly consistent between ascending and descending orbits, with minor discrepancies potentially explained by differences in the LOS and acquisition time (18:00 local for ascending, 06:00 local for descending) of the two relative orbits.

Applying our technique to identify and map areas of buried ice, we observed a substantial decrease (0.34) in median coherence over the moraine dam moving area between the winter–early spring (0.92) and late summer (0.58) (Fig. 5). There was no measurable decrease over the stable area.

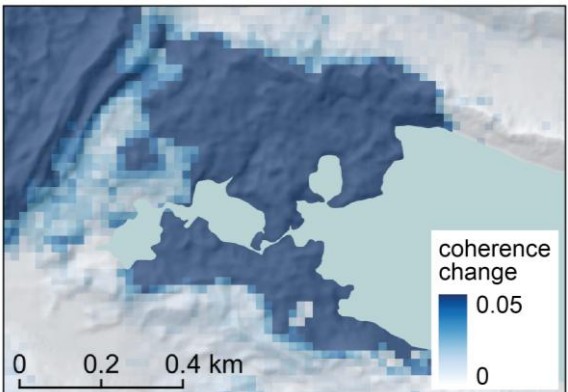

**Figure 5: Seasonal change in median coherence offers a proxy to map the spatial distribution of buried ice.**

## 5.2 Feature-tracking offset maps

The feature-tracking offset maps showed coherent motion of the moraine dam surface. The spatial mean of the per-pixel median LOS velocity over the moving area was +13.6 cm/yr for the ascending orbit and +7.8 cm/yr for the descending orbit, with a maximum per-pixel median LOS velocity of +46.1 cm/yr for the ascending orbit and +29.3 cm/yr for the descending orbit (Fig. 6A, B). For both orbits, higher velocity was observed on the eastern side of the moving area near the lake shoreline. The mean SNR of the feature tracking products over the moraine dam moving area was 11.4 for the ascending orbit and 14.0 for the descending orbit (Fig. 6C, D). The spatial mean of the per-pixel standard deviation of LOS velocity was 13.7 cm/yr for

the ascending orbit and 25.9 cm/yr for the descending orbit (Fig. 6E, F). The spatial mean of per-pixel median LOS velocity over the stable area was 0.7 cm/yr for the ascending track and -0.3 cm/yr for the descending track, though the feature tracking velocity values were ultimately masked in these areas with high InSAR coherence prior to inversion (Section 4.2). Positive LOS velocities are also observed over the Lhotse Glacier northwest of the moraine dam (Fig. 6A, B).

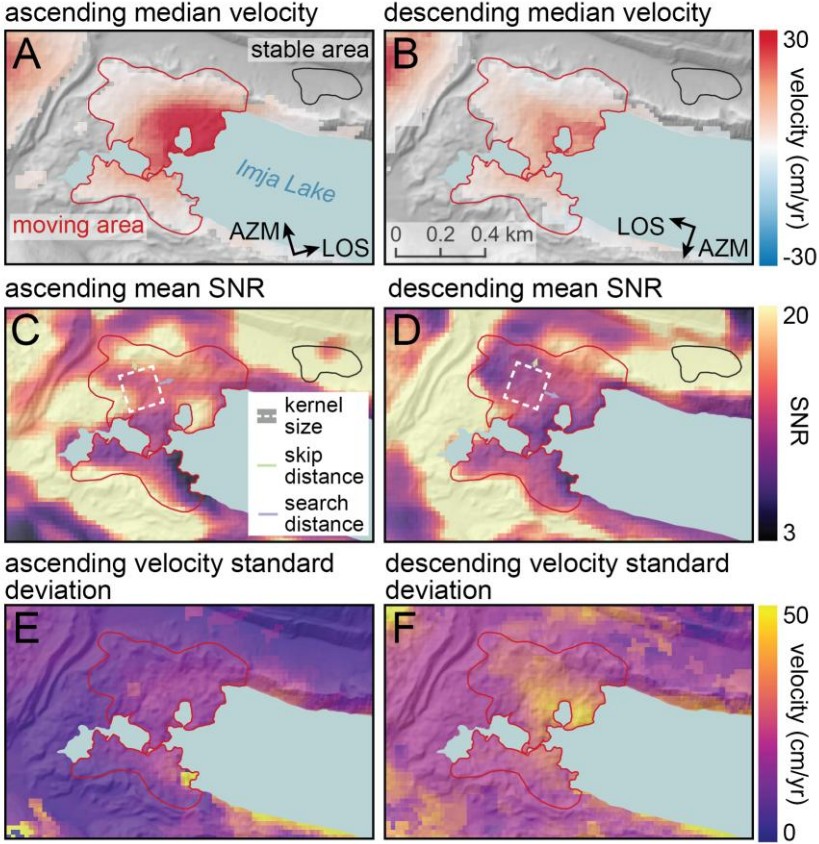

**Figure 6: Sentinel-1 SAR image feature tracking results for the full 2017-2024 time series, including all 192 pairs on ascending (left) and 210 pairs on descending (right) orbits. A, B) Per-pixel median feature tracking velocity in the slant-range direction (along the line-of-sight). Results are masked in areas with high InSAR coherence (Figure 4), where feature tracking products are not used during inversion. C, D) Per-pixel median signal-to-noise (SNR) ratio. E, F) Per-pixel standard deviation of the feature tracking**
**velocity in the range direction. Higher standard deviation values are observed for locations with 1) greater temporal variability, 2) fewer reliable displacement measurements, and/or 3) larger measurement error.**

### 5.3 Cumulative displacement time series from combined InSAR and feature tracking inversion

Most pixels in the moving area showed positive range change (i.e., movement away from the satellite) over the course of the study period for both ascending and descending orbits (Fig. 7A, B). The mean total LOS displacement of the moraine dam
moving area over the 7-year study period was 97.7 cm for the ascending orbit and 50.5 cm for the descending orbit, with a maximum LOS displacement of 338.6 cm for the ascending orbit and 207.4 cm for the descending orbit. The largest displacements were observed on the east side of the moving area near the lake shore, while limited displacement was observed

on the northern, southern, and western margins of the moving area. While many pixels displayed constant LOS velocities over time, others showed apparent seasonal fluctuations (Fig. 7E, F). To assess error caused by atmospheric noise, we first computed the mean apparent displacement in the stable area at each time step, then computed the standard deviation over all time steps. These values were 0.2 cm/yr for the ascending orbit and 0.3 cm/yr for the descending orbit, indicating that our approach for atmospheric noise removal was effective.

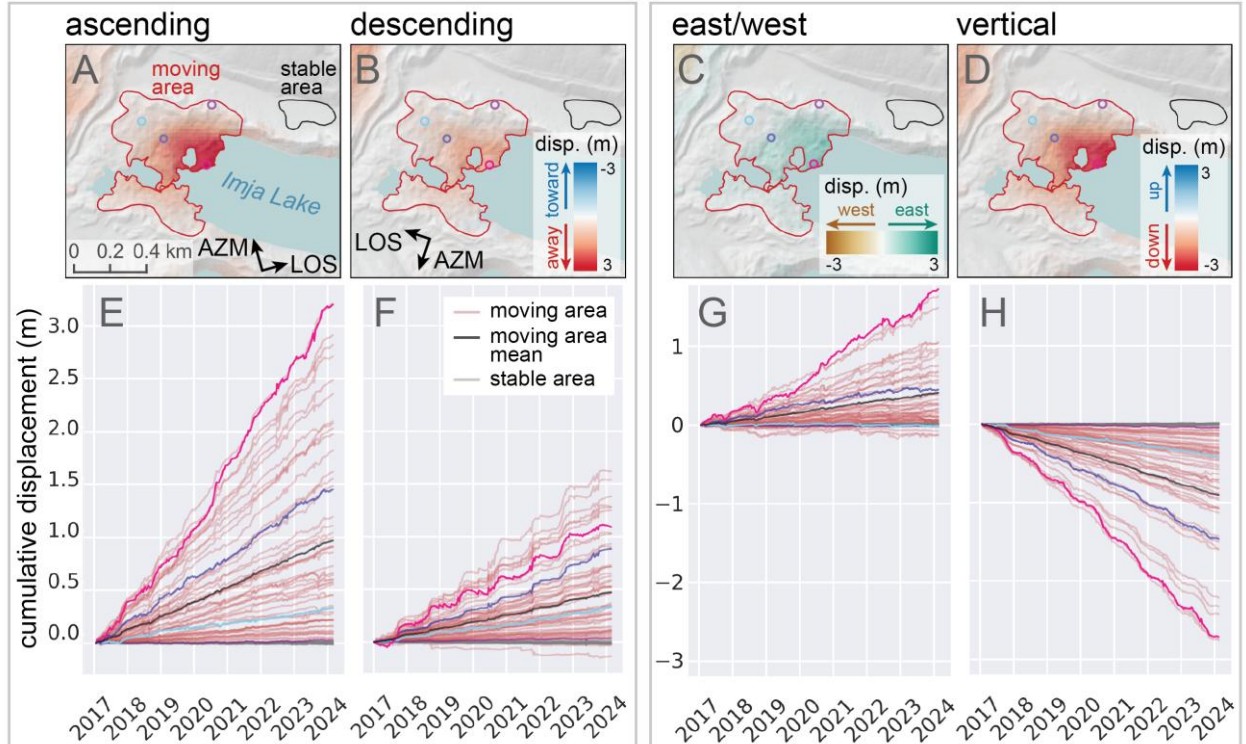

**Figure 7: Cumulative displacement time series from the combined InSAR and feature tracking inversion. A–D) Maps showing total cumulative line-of-sight (LOS) displacement of the moraine dam and stable area over the study period for the ascending (A) and descending (B) orbits, and the decomposed east/west (C) and vertical (D) directions. Colored circles show pixel locations for the 1st, 25th, 75th, and 99th percentile of cumulative ascending displacement values in the moving area. E–H). Cumulative displacement time series for 50 randomly selected pixels in the moving area (red lines) and 50 randomly selected pixels in the stable area (grey lines). The black lines show the mean of all pixels in the moving area. The colored lines show the cumulative displacement time series for the pixels marked with corresponding colored circles in the maps.**

After decomposing the LOS displacements into vertical and east/west displacements, we found that the observed LOS displacement of the moraine dam is consistent with downward and eastward motion (Fig. 7, 8). The mean total vertical displacement of the moraine dam moving area over the study period was -93.4 cm, with a maximum total vertical displacement of -337.0 cm. The mean total eastward displacement was 38.3 cm, with a maximum displacement of 174.1 cm. Over the stable area, the mean total vertical and horizontal displacements were 0.7 cm and 0.8 cm, respectively. While some pixels showed

constant velocity over time, the pixels with the highest velocity displayed more seasonal variability, with increased velocity during the warmer months (Fig. 7G, H).

The monthly median velocity maps show year-round movement of the moraine dam with a clear seasonal cycle. Vertical velocity was lowest (between -8.6 cm/yr and -10.1 cm/yr) between January and April (Fig. 8A, B). The moraine dam moving area vertical velocity increased from May to October, when the mean vertical velocity reached -21.7 cm/yr (Fig. 8A, B). The vertical velocity over the moraine dam moving area decreased in November and December (Fig. 8A).

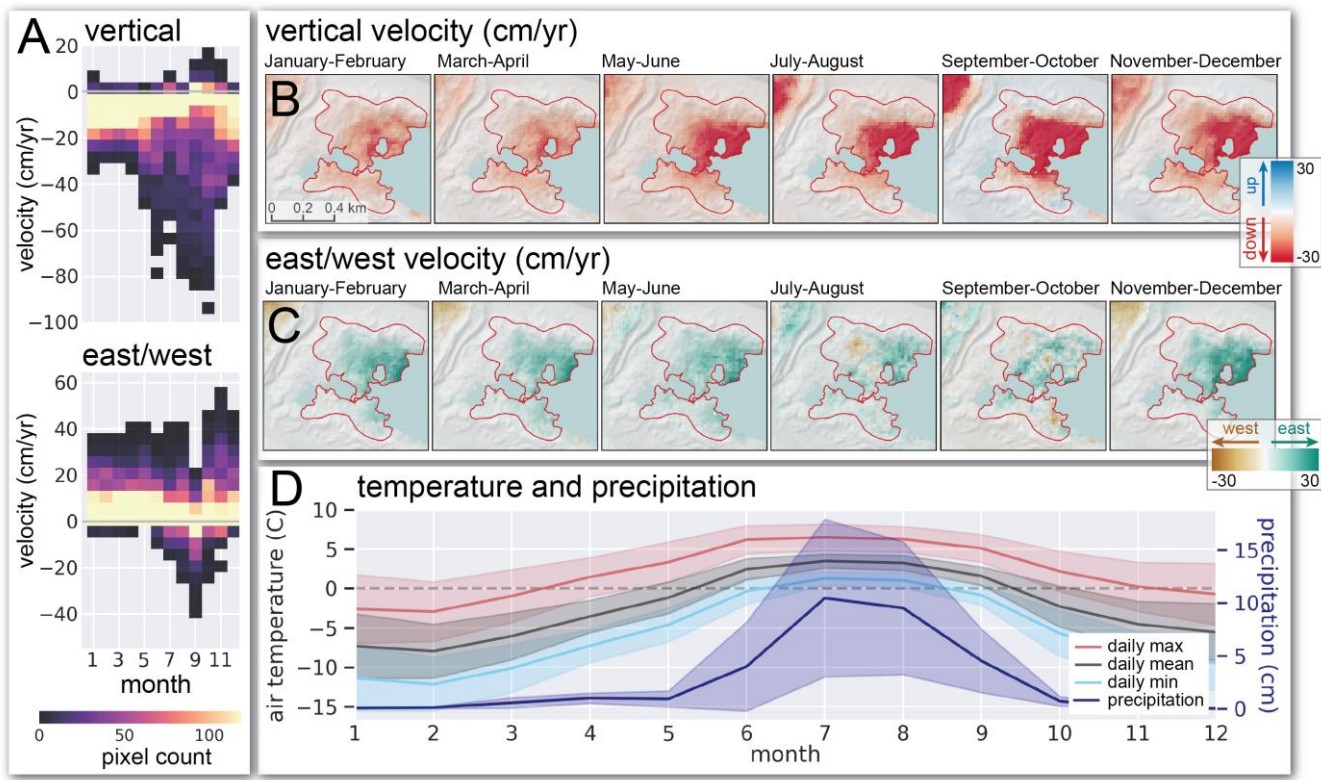

**Figure 8: Aggregated monthly surface velocity of the Imja Lake moraine dam from the combined InSAR and feature tracking inversion for the 2017 to 2024 study period. A) Histograms showing distribution of median monthly surface velocity values in the moving area. B, C) Maps showing vertical and east/west median surface velocity during bimonthly periods. D) Mean daily air temperature and precipitation for each month at the nearby Pyramid Weather Station (UCAR/NCAR-Earth Observing Laboratory et al., 2011). Shaded areas show one standard deviation.**

From December to May, the mean horizontal velocity of the moraine dam moving area was 6.0 to 8.3 cm/yr to the east, with limited pixel-to-pixel variability (Fig. 8A, C). Between June and September, some pixels began to move westward, and pixel-to-pixel variability increased (Fig. 8A, C). The mean east/west velocity in the moving area decreased from June through September (Fig. 8A, C).

### 5.3 Validation

Our analysis shows that both the magnitude and direction of the combined InSAR and feature tracking velocity time series are consistent with the corresponding DEM-derived validation components (Fig. 9; Table 2). The mean vertical velocity in the moving area from DEM differencing was -15.0 cm/yr. The mean vertical velocity from our combined InSAR and feature-tracking time series approach was -13.0 cm/yr. The InSAR-only SBAS underestimates both vertical and horizontal velocity (Fig. 9). The feature-tracking-only SBAS captures velocity magnitude, but fails to capture short-term variability and contains

large errors.

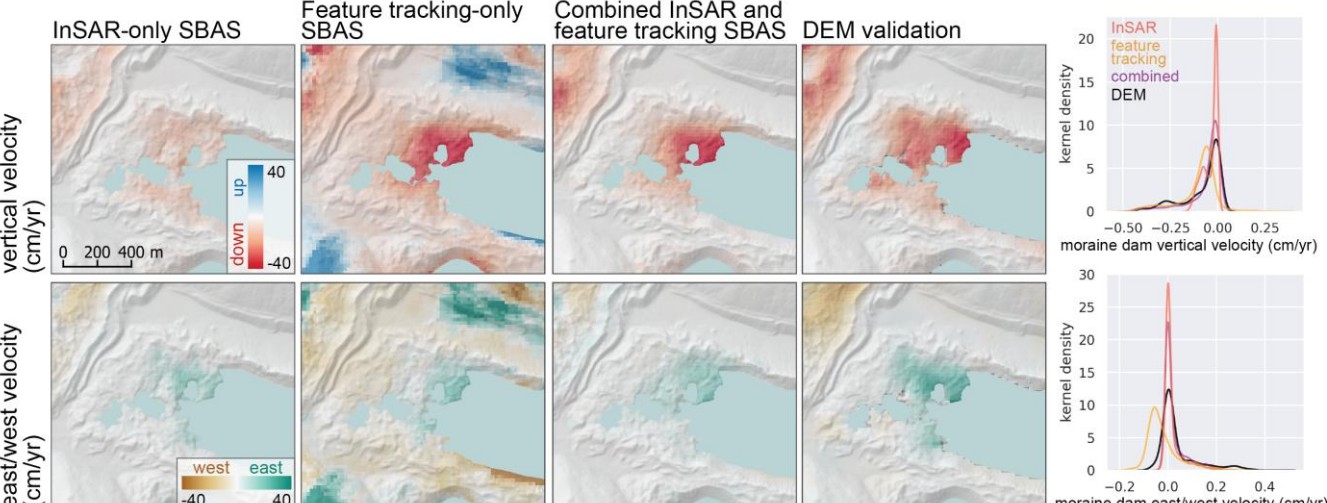

**Figure 9: Mean vertical (top) and east/west (bottom) velocity products from InSAR-only SBAS (left), feature-tracking-only SBAS (left center), our combined InSAR and SAR feature tracking SBAS approach (right center), and the downsampled DEM-derived validation data (right), with corresponding kernel density plots for the moraine dam moving area. Our combined approach provides**
**the best agreement with the validation data.**

**Table 2. Moraine dam velocity components (up/down and east/west) from InSAR-only SBAS, feature-tracking-only SBAS, our combined InSAR and feature tracking SBAS approach, and the DEM-derived validation data. The $R^2$ values were computed from per-pixel differences of each approach with the corresponding DEM-derived validation data.**

| | Mean up/down velocity (cm/yr) | Mean east/west velocity (cm/yr) | Std up/down velocity (cm/yr) | Std east/west velocity (cm/yr) | $R^2$ up/down velocity | $R^2$ east/west velocity |
|---|---|---|---|---|---|---|
| InSAR-only SBAS | -3.7 | 1.8 | 3.9 | 3.9 | 0.63 | 0.82 |
| Feature-tracking-only SBAS | -10.2 | 2.0 | 10.6 | 6.9 | 0.67 | 0.35 |
| Combined SBAS | -6.7 | 2.4 | 9.9 | 4.6 | 0.78 | 0.58 |
| DEM-derived validation | -8.0 | 4.0 | 10.6 | 8.1 | – | – |

 **6 Discussion**

Our surface displacement and coherence change results can be used to both infer the primary processes responsible for moraine dam degradation and to constrain the timing and magnitude of those processes.

**6.1 Ice melt**

We interpret the observed seasonal decrease in coherence to be caused by melting of buried ice and associated debris reworking. The lag between the onset of air temperatures above freezing and melt onset (Fig. 4) is likely related to debris cover insulation. Both the melting of buried ice and debris reworking will change the moraine surface geometry, altering the scattering characteristics of the surface and decreasing coherence.

Where the magnitude of the seasonal change in coherence is large (Fig. 5), we assume that buried ice is present. We find evidence for buried ice over most of the moraine dam, including areas displaying near-zero surface displacement. This is consistent with previous geophysical survey results. Hambrey et al. (2008) used both GPR and ERT to document buried glacier ice tens of meters thick (over 40 meters in places) overlain by up to 20 m of debris along the eastern portion of the moraine. Somos-Valenzuela et al. (2012) found consistent evidence for buried ice throughout 13 GPR transects across the moraine dam, ranging in bottom depth from 0–65 m. The deepest ice bottom depths (40–60 m) were documented on the northeast moraine dam, while shallower ice bottom depths (20–40 m) were documented around the southwest margin of the moraine dam, near the outlet. Dahal et al. (2018) conducted a dense ERT survey of the area surrounding the moraine dam outlet to the southwest, finding evidence for discontinuous blocks of glacier ice at depths varying from 0–20 m. Note that Hambrey et al. (2008) and Dahal et al. (2018) estimated debris thickness but were not able to locate the bottom of the debris-covered ice, and thus presented only a lower bound on ice thickness. Meanwhile Somos-Valenzuela et al. (2012) presented an ice-bottom depth, but did not estimate debris thickness, resulting in only an upper bound on ice thickness. We observe a smaller seasonal change in coherence over the southwest corner of the dam near the outlet (Fig. 5), which is consistent with the isolated ice deposits mapped by Dahal et al. (2018). The magnitude of observed seasonal coherence change is likely also related to overlying debris thickness, as thicker debris 1) provides more insulation, suppressing melt, and 2) experiences less surface change per volume of ice melt at depth.

While other processes could also contribute to decreased coherence during warmer months, we did not observe loss of coherence over the stable area during the warmer months (Fig. 4D, H). This suggests that seasonal melt of massive, buried ice is primarily responsible for the observed coherence loss over the moraine dam.

The observed spatial variability in seasonal coherence change may be explained by debris thickness and proximity to the lake and surface ponds, which can enhance downwasting and backwasting (Johnson, 1971; Driscoll, 1980; Watanabe et al., 1995).

Low coherence during colder months was observed in some locations, especially bordering the lake and surface ponds (Fig. 4A, C). These areas tend to coincide with areas experiencing rapid subsidence in our DEM-derived validation data (Fig. 3). When air temperatures are below freezing, these locations potentially experience additional ice degradation and associated debris reworking due to melt caused by contact with water and by calving.

Overall, our measured vertical velocity values (Fig. 7, 8; Table 2) are lower than those documented by earlier studies. Watanabe et al. (1995) measured the position of five painted boulders in November 1989 and October 1994 and documented cumulative vertical displacements of -0.3 to -13.5 m with a mean of -4.8 m over the 5-year period (corresponding to vertical velocity range of -0.06 to -2.70 m/yr and mean of -1.0 m/yr). Fujita et al. (2009) performed topographic surveys using a theodolite in November 2001 and GPS receivers in April 2002 and October 2007. They created digital elevation models (DEMs) with 1 m posting for each survey and differenced to estimate cumulative vertical surface displacement. They report cumulative vertical displacement values for the "left bank" and "right bank" of -1.63 ± 1.71 m and -1.97 ± 1.67 m (mean ± 1-sigma), respectively, with a total range of vertical velocity values over the moraine dam of -0.06 to -1.03 m/yr for the 6-year period from 2001 to 2007. Haritashya et al. (2018) measured mean vertical velocity over the moraine dam between February 2000 and February 2016 of less than -0.5 m/yr, with a maximum of -1.9 m/yr, and uncertainty of 0.7 m/yr. The disagreement with our measured vertical velocity values likely results from some combination of inaccurate and/or incomplete sampling of the moraine dam surface in previous work, and/or a real decrease in the moraine dam vertical velocity over time.

During the warmer months, we observed a clear increase in downward vertical velocity over much of the moraine dam (Fig. 8A, B), likely associated with melting of buried ice. The highest downward velocity over the moving area was observed in October with mean velocity of -21 cm/yr. Seasonal increases in velocity were first observed over the northern margin of the lake and surface ponds, suggesting earlier seasonal melt in these locations. Small areas of apparent upward displacement at the southern margin of the lake (Fig. 8B) are likely caused by consistent unwrapping errors due to low coherence (Fig. 4B, D). East-west displacement during the warmer months was variable, with some areas mostly moving east and some areas mostly moving west (Fig. 8C). This variability in horizontal displacement direction could be caused by the variable topography of the moraine dam, as the apparent direction of displacement caused by melt of sloped ice surfaces depends on the surface aspect.

If we extrapolate the observed mean subsidence rates for the 2017 to 2024 period forward in time and consider the current moraine dam topography (Fig. 1, S1), we find that the northeast area of the moraine dam bordering the lake edge is most vulnerable to subsidence below the current lake level. This topographic evolution would result in westward expansion of Imja Lake, decreasing the width of the moraine dam.

## 6.2 Ice flow

During the colder months, we observed widespread downward and eastward displacement of the moraine dam surface, with a
mean velocity within the moving area of -10.5 and 7.3 cm/yr, respectively (Fig. 8). The high coherence values over the moraine
dam during colder months (Fig. 4) provide high confidence in these measurements. Assuming that air temperatures during the
colder months were mostly below freezing (Fig. 4C, G), we do not attribute this displacement to melt of buried ice. While
debris settling and frost creep could potentially explain some component of downward and horizontal motion, they are not
expected to cause displacement of this magnitude on these time scales (Colman and Pierce, 1986; Bursik, 1991; Hallet and
Putkonen, 1994) and we do not observe similar motion over debris outside the moraine dam. We instead attribute this motion
primarily to flow of the thick, massive ice within the moraine dam.

Downward and eastward movement is consistent with along-slope flow for the broad moraine dam surface slopes (Fig. S1).
The mean surface slope of the moving area is 9.5˚. Somos-Valenzuela et al. (2012) found evidence for ice bottom depths
between 30 and 60 m over much of the moraine dam. Assuming debris thickness of less than 1–20 m (Hambrey et al., 2008),
buried ice in the moraine dam should have thickness of 10 to 60 m. We generally observe higher east/west and vertical surface
velocity during colder months over areas with deeper ice bottom depths documented by earlier studies. Near-zero velocity was
observed over the southwest portion of the moraine dam, where Dahal et al. (2018) observed discontinuous blocks of buried
ice. The observed mean winter velocity magnitude (combining the vertical and east/west components) of 12.8 cm/yr over the
moving area of the moraine dam is consistent with expected surface velocity values for a simple ice flow model and the range
of measured ice thickness (See supplementary Section S1, Fig. S9).

Previous surface displacement measurements also provide evidence for flow of buried ice within the moraine dam. Watanabe
et al. (1995) described westward movement of several boulders along the outlet channel of -0.96 m/yr. Fujita et al. (2009)
described horizontal flow rates of less than 0.17 m/yr in an unspecified direction. These results potentially suggest that at some
point before our study period, the flow direction of the moraine dam evolved from original westward, downvalley flow (the
original flow direction of Imja Glacier) to the eastward, upvalley flow (toward Imja Lake) observed during our 2017–2024
study period.

The observed change in flow direction is likely related to evolving thickness of lower Imja Glacier through differential ablation
both above and below the lake surface (Kjær and Krüger, 2001), changing longitudinal stresses associated with decoupling
from the active flowing glacier during retreat, and interaction with the expanding lake. We observed a concave-upward
transverse profile (bottom-right panel of Fig. S10) for the Imja Lake moraine dam during our study period. A similar
configuration, where the glacier surface slopes downward from the lateral and terminal moraines toward the centerline, is
observed for many debris-covered glaciers in the Everest region (e.g., Nuptse Glacier, Fig. S11). This scenario introduces

driving stress toward the glacier centerline (downslope), which is consistent with the observed direction of motion in our horizontal surface velocity maps. Our detailed displacement time series can potentially be used to constrain ice and debris evolution models, which are needed to better understand the evolution of the Imja Lake moraine dam and implications for changing structural stability.

## 6.3 Limitations

While our InSAR and feature tracking results improve our understanding of the processes responsible for moraine dam degradation, it is important to discuss some limitations. Underestimation of rapid, localized surface displacement can partly explain why our maximum vertical velocity over the moraine dam surface was only -0.47 m/yr compared to -1.9 m/yr from Haritashya et al. (2018). We also see evidence of localized backwasting in our DEM-derived validation dataset (Fig. 3) that is not captured in our InSAR and feature-tracking displacement time series (Fig. 7, 8).

We also acknowledge limitations of the Sentinel-1 acquisition geometry, with only ~10% of any real north/south displacement mapped into LOS measurements for this location. We combined data from ascending and descending passes to calculate vertical and east/west displacement components, assuming that north/south displacement does not substantially contribute to the displacement observations. In fact, some small contribution from north/south displacement is expected, which potentially introduces a source of error in our measurements of surface velocity (Brouwer and Hanssen, 2021). Based on the north/south displacement from the DEM-derived validation dataset, we expect that our LOS decomposition could introduce a bias of less than ~4% in mean vertical velocity over the moraine dam moving area and less than ~3% in mean east/west velocity over the moraine dam moving area (Section S2, Fig. S12, S13).

## 7 Conclusions

We demonstrated that Sentinel-1 InSAR and SAR feature tracking can be used to measure the displacement of an ice-cored moraine dam bounding Imja Lake in Nepal. We validated our measurements using a high-resolution 3D displacement dataset derived from stereo DEMs. By combining InSAR and feature tracking measurements, we created a surface displacement time series that is more accurate than those created using InSAR or feature tracking alone. We found that a 0.3 km$^2$ area of the moraine dam subsided an average of ~90 cm over the 7-year study period, and the northeastern area of the moraine dam bordering the lake edge is most vulnerable to subsidence below the current lake level.

We observed a systematic seasonal cycle in InSAR coherence values over the moraine dam surface, with a substantial decrease in coherence occurring about 60 days after mean daily maximum air temperature increased above 0°C. We showed that seasonal changes in InSAR coherence can be used to map buried ice within the moraine dam, with an observed spatial distribution that is consistent with previous geophysical surveys.

Our results suggest that the primary processes contributing to moraine dam degradation are buried ice melt and ice flow. Buried ice melt occurs in the warmer months (summer and fall), causing a seasonal increase in downward vertical velocity and variable horizontal velocity over the surface of the moraine dam. Ice flow occurs year-round and results mostly in eastward movement. Our surface velocity measurements, alongside historical measurements, provide evidence that the Imja Lake moraine dam is undergoing kinematic changes, which we attribute to changing ice thickness, past decoupling from the Imja Glacier, and interaction with Imja Lake.

Collectively, our findings demonstrate the potential for satellite SAR remote sensing to quantify the evolution of moraine dam degradation and to provide new information about moraine dam internal structure. This information may be valuable when assessing GLOF hazards, performing in situ investigations, and planning hazard remediation activities. We are extending this analysis to other high-priority moraine dams in the region and hope to integrate additional remote sensing measurements of surface change (e.g., NISAR, optical feature tracking) in the future.

**Code availability**

All source code and documentation are available on Github. The fufiters repository (https://github.com/relativeorbit/fufiters) contains code and documentation for reproducible data processing and the fufiters_imja_analysis (https://github.com/gbrencher/fufiters_imja_analysis) repository contains code for data analysis and figure preparation. The versions of this code used to prepare the materials for the published manuscript are archived on Zenodo (Henderson and Brencher (2025) and Brencher (2025), respectively).

**Data Availability**

The InSAR coherence time series, cumulative displacement time series, and DEM-derived validation data prepared for this study are available on Zenodo (Brencher et al., 2025). All Sentinel-1 data are freely available from the Alaska Satellite Facility (ASF) Distributed Active Archive Center (DAAC). The Maxar WorldView Level-1B images used in this study are available under the NGA NextView and Electro-Optical Commercial Layer (EOCL) licenses.

**Author Contribution**

GB conceptualized this work with input from DS. GB and SH designed the methodology and developed the software. DS prepared the validation dataset. GB performed the analysis and prepared the manuscript with substantial input from DS and SH.

## Competing Interests

The authors declare that they have no conflict of interest.

## Acknowledgements

GB was supported by NSF GRFP DGe-2140004. Analysis was performed using CryoCloud (NASA grants 80NSSC22K1877, 80NSSC23K0002). DS was supported by NASA awards 80NSSC20K1595 and 80NSSC24K1633. SH was supported by NSF award 2117834 and NASA award 80NSSC22K0345. We thank the two anonymous reviewers and Tobias Bolch for their insightful, constructive comments, which improved the manuscript. We thank Eric Gagliano, Alexander Handwerger, Alison Duvall, and Bart Nijssen for helpful discussions about this work.

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
