# Peer review of "Quantifying degradation of the Imja Lake moraine dam with fused InSAR and SAR feature tracking time series"

_EGUsphere, 2024_

## Referee Comment (RC1)

**Quantifying degradation of the Imja Lake moraine dam with fused InSAR and SAR feature tracking time series**

Brencher et al., 2024
* * *
The authors describe a very interesting study using a combination of InSAR and SAR feature tracking to document the degradation of the Imja lake moraine dam. Their work offers valuable insights into ongoing changes, which can be useful for hazard assessment. Generally I find the study carefully executed and the results are largely convincing. To better convey the message, however, I believe that the manuscript would benefit from some mild restructuring and occasional rewording. Below, I provide some general comments and a number of specific recommendations that I believe will address these shortcomings.

**General comments**

- Given the focus of the group within which this work was performed, I am surprised that no DEM differencing was performed. Given the main findings (ice melt / surface lowering), it seems like DEM differencing would be the an important source of information? Personally, I would be delighted to see the manuscript supplemented with this information, but I recognize that this is a big ask. In any case, it would be interesting to say why this could not or was chosen not to be done.
- I think the discussion falls a bit short of expectations. It nicely discusses the study results, but I think it would deserve a slightly boarder view on how and why this information matters. I have included a number of questions that the authors could consider addressing (more as a menu than an expectation that they are all addressed, and I am sure the list is not comprehensive).
- The manuscript is generally well written, but frequently uses very short paragraphs, which can disrupt the reading. I recommend combining some of these in such a way that a paragraph is at least 3-5 sentences (and less than half a page). For my taste subsections in the intro are not needed (use carefully crafted topic sentences instead). A clear statement of the aim of the paper and the chosen approach to address the knowledge gap are missing in the introduction / the information is scattered throughout, consider tightening. What is true for paragraphs is also true for subsections (e.g., 4.4. and 4.5 )
  A note on literature review statements: If you avoid having the citation at the start of the sentence – effectively making it the grammatical subject of the sentence – the relevant information becomes more apparent. Compare the following:
  *Hambrey et al. (2008) used both GPR and ERT to document buried glacier ice tens of meters thick (over 40 meters in places) overlain by up to 20 m of debris along the eastern portion of the moraine.*
  vs.
  *Thick glacier ice – over 40 m in places and overlain with up to 20 m of debris – has been documented along the eastern portion of the moraine (Hambrey et al., 2008*)  [I removed the reference to GPR and ERT because you already stated this in a previous sentence]

**Specific comments**
L36: Consider mentioning early warning systems in addition to engineering solutions? A good reference could be Niggli, L., Allen, S., Frey, H., Huggel, C., Petrakov, D., Raimbekova, Z., Reynolds, J., Wang, W., 2024. GLOF Risk Management Experiences and Options: A Global Overview, in: Oxford

Research Encyclopedia of Natural Hazard Science. Oxford University Press. https://doi.org/10.1093/acrefore/9780199389407.013.540

L39: You jump directly to "moraine dams" here, but might want to introduce to reader first to the different types of dams that exist? A good reference to include would be Rick, B., McGrath, D., Armstrong, W., McCoy, S.W., 2022. Dam type and lake location characterize ice-marginal lake area change in Alaska and NW Canada between 1984 and 2019. The Cryosphere 16, 297–314. https://doi.org/10.5194/tc-16-297-2022

L40: Add strong precipitation events as possible trigger?

L44: It's not clear here whether you imply that the evolution of the moraine dam reduces or increases the GLOF likelihood (I guess only lowering is implied here?). Both are plausible in my mind, maybe specify?

L64: It seems relatively uncommon to reference Fig. 1 already in the introduction. I don't think it's necessary, and it makes the reader's life unnecessarily hard when trying to refer to the figure during the "study site" section. Consider moving it and the first reference to it to section 2.

L105: rotational failures (does it have to be rotational?) into the lake or on the other side of the dam? Either it could be a trigger for a GLOF or simply weaken the moraine. Maybe specify? You could also reference the South Lhonak Lake event.

L114: I think there is a conceptual error here, since feature tracking velocities are not along the line of sight.

L116: My understanding is that measuring flow velocities of glaciers is/was only possible with quite some uncertainty and only in specific locations. Maybe specify a bit more in detail when this is possible (and when not!). Saying that this can be done with mm-level precision is likely overselling it a bit.

L129: Consider adding reference to Jacquemart, M., Tiampo, K., 2021. Leveraging time series analysis of radar coherence and normalized difference vegetation index ratios to characterize pre-failure activity of the Mud Creek landslide, California. Natural Hazards and Earth System Sciences 21, 629–642. https://doi.org/10.5194/nhess-21-629-2021 for the coherence.

L153: I found myself wondering if the rate holds true for the period after the publication of the reference given here (Watanabe 2009), since almost 20 years have passed? Do you have a sense of how constant it has been over time?

L163ff: It would be neat to plot the ERT / GPR lines of other studies (color coded by ice depth?) in Figure 1 if you can get your hands on the data. It would greatly facilitate the description of the ice (thickness) distribution. In one case you refer to "ice bottom depth" and in another place simply "40 m thick". Do these values mean the same?

L167: Can debris in the dam really be fully saturated while retaining its stability?

L174ff: I would have expected these detailed values in the discussion, rather than in the study site. I think for the description of the study site, a brief overview of where what kind of motion has been measured would suffice and be easier to remember (e.g., this corner moved most in this direction, this other part didn't move much etc.) Decide on GNSS or GPS (Technically these should also be defined). Not clear who "the authors" are in the last sentences.

L197: consider supplementing "azimuth" and "range" with along-flight and across-flight directions such that these are defined here for future use and comprehensibility for non-radar experts.

L216: consider simply saying (or specifying) that a *downsampling* factor of 5 was used in the range direction while the native resolution was kept for the azimuth. The "looks" terminology is very radar specific.

L221: how high did you chose your mean coherence for the reference and stable areas? What is the difference between the "reference" and "stable" area? Why are they different?

L237: I presume you mean horizontal velocity here?

L245: the feature tracking is not along the line-of-sight (but along slant-range and azimuth direction).

L250: over what period is the median? The whole stack or the subset of pairs based on the temporal baselines?

L254ff: I am confused about this procedure. I am probably just misunderstanding, but here's what trips me up: First, you speak about converting the feature tracking offsets to radians. Why do you do/need this? Is it required by MintPy? Then you mask pixels with a SNR lower than 8 (how/where was the SNR calculated, and why do you choose this value?), then you remove pixels with displacement more than 10 m. But I thought you had converted the displacements to radians? So do you do the filtering first? Why do you remove measurements with a high InSAR coherence? Wouldn't the overlapping measurements provide additional robustness to your findings? I think some more details would be helpful here.

Section 4.2: You could consider adding a flow chart or other graphical representation of all the steps in your processing. Currently it's a bit hard to follow. This would also benefit from a geometric diagram, showing the LOS, the azimuth and (slant) range directions, since these can be quite confusing for non radar-experts.

L268: A threshold of 0.6 is quite high, why did you chose it?

L270: I don't understand how or to what the per-pixel median feature tracking velocity products were scaled.

L273: What were the per-pixel displacement measurements inverted to, I don't fully understand.

Sec. 4.3: Did you do all this in MintPy or with code of your own? My understanding is that MintPy is able to do this, no?

Sec. 4.4: Doesn't this just belong to Sec. 4.3? I would combine ☺

Sec. 4.5: This information seems incomplete. What did you then do with these coherence maps? Did you visually compare where there were changes and mapped these? Did you use some kind of threshold? How did you deal with snow decreasing the coherence in winter? Can you specify what months the maps cover? Is it the same as what you present in Fig. 2?

Sec.5.1/Fig.2:
- You list 8 temporal baselines, but I can only count 5, maybe 6 in the figure. Consider removing the ones that are not in the figure, if they are indeed missing?
- I would also recommend changing the color scale of plots C,G,D, and H to something different from panels A,B,E,F because they show very different quantities, and it is initially very confusing (e.g., you could use viridis for one and magma for the other).
- You outlined the area with lower coherence at the front of the lake, but the moraines along the lake also show up as low coherence areas. Why did you exclude those from the "moving area"? Is something else causing the low coherence area there? Or are they also ice-cored? (→ address in the discussion)

Sec.5.2/Fig.3:
- I believe (as highlighted elsewhere, sorry for the repetitive comments) that the LOS you mention in connection with the feature tracking is actually the slant-range. Replace throughout.
- I am a little bit perplexed as to why the velocities from your feature tracking show positive velocities in both orbits. How do you interpret this? Combined with the rather low SNR over this whole area, do you believe that these are indeed horizontal displacements? Or do you think they are in fact a component of large-scale subsidence? Without getting too discussion-y, it would be nice to give a bit more context here.
- I ask this also because the same signal appears on Lhotse glacier, in the top-right corner of your panels. On that note, somewhere in the manuscript (maybe in the study section?) it would be nice to point out that there is another glacier there, which also shows a signal. That way, the readers can understand those spatial patterns.

Sec. 5.3/Fig.4:
- What I miss between section 5.2 and 5.3 is a comparison / context regarding the different magnitudes of motion. In 5.2 you describe the results in terms of velocities in cm/year, and then

in 5.3 you provide total displacements. For me, these are hard to reconcile in my head. Can you somehow standardize?

- Between sections 5.2 and 5.3, the reader is provided with many different quantities and numbers, but it's relatively hard to do much with this information. At the end of section 5.3, you provide an overarching interpretation of the combined patterns. My suggestion would be to put that information upfront and say something like: "The observed patterns of deformation are consistent with downward and eastward motion of the moraine, as is evidenced by the feature tracking and InSAR inversion. The feature tracking shows …"
This might even remove the requirement of having the two subsections 5.2 and 5.3.
- On a related note, I was somehow surprised not to see the results from the InSAR on its own. Can you provide these or justify in the text why you don't see the need to do so? Personally, I would very much like to see a few wrapped interferograms to judge the data quality coming from the InSAR.
- Fig.4 E-H: Is this the combined cumulative displacement or just from InSAR? The label on the y-axis should probably be cumulative displacement. Can you additionally mark (in the maps and in the profile lines) a few more points (e.g., 2 above and 2 below the mean? That would provide a bit of insight into the spatial patterns and lend credibility to your inversion.
- Fig. 4 A-D: Personally, I would use a different colormap (e.g., magma) for the vertical displacement because you are showing a new quantity. I would also appreciate a slightly more permissive color range (+/- 3m?) to show more small scale patterns.

Sec. 5.4:
- L380: For my taste you don't need to reiterate the velocities of the stable area here, since you clearly made the point that they are negligible.
- Try to avoid "noun trains" (e.g. moraine dam moving area vertical velocity"). Why not say "the vertical velocity of the moraine dam increased from May to October, when it reached a maximum of …"
- Rather than speaking of "vertical velocity" (here and earlier on, but here at the latest), why don't you call it subsidence? This would be more clear.
- Fig.5: This is a detail, but I am personally bothered by the "plastered on" scale bars ;-) Can you put them next to each other (both horizontal) between panels B and C?
- I don't believe that section 5.4 deserves its own subsection, since it's just another result of the combined inversion. I would integrate with the other part(s).

Sec. 5.5
- This section (2 sentences!) is a bit comical ;-). Since this comes directly from the coherence analysis, why do you not just integrate it with section 5.1? It's the obvious interpretation of your coherence analysis, and lays the perfect justification for why you primarily detect downward motion with your other analyses.

Discussion:

L442: Could you compare the values for the ice flow during the cold months to what would be expected for a glacier from only the internal deformation / creep component (Cuffey and Patterson, sec. 8.3, I think)? Are they anywhere in the same range?
L468: Can you speculate about why the flow direction may have reversed? Did the overall slope change since the early 1990s?
L471: Some confusion here: are you saying that the direction of the ice flow mentioned above (L466) was actually the direction of the ice flow of the original glacier? Or were these measured post separation?

L475: Are you saying here that Imja Lake is actually still a supraglacial lake? Or what do you mean by "complete decoupling from the active glacier will remove compressive stresses"? Specify clearly (and here in the Study Site, for I certainly missed this if it is at all addressed there).
Sec. 6.3: As elsewhere, I would integrate these findings with the discussion of the subsidence.

Questions that I would like / was expecting to see addressed in the discussion:
- How does the subsidence / ice melt compare to measurements from other places? Can you estimate from this how long you expect the ice core to stick around? What will Imja lake look like, once it's all gone? How does it affect the hazard?
- What implications do you think these kinds of measurements can have for operational monitoring. What drawbacks or caveats?
- Under what circumstances do you think your approach can work (in terms of geometry of the moraine/valley)? Would this work on most glacial lakes in HMA or is there something special about this one (on that note, a justification about why you chose Imja in the Study Site description would be nice).
- All the numbers presented in the Study Site (with regards to the changes measured by others) should be presented in the discussion, I think, giving context and comparability of your results. If possible, it would be really cool to put all the different numbers on map.

S6.4: It unfortunate to end your strong manuscript with a long discussion of the limitations, because you leave the reader with information that suddenly calls into question all the work you just presented. As a reader, it requires me to re-evaluate what I just read and put the interpretations in light of the limitations I am only provided with now. I would recommend moving the different parts of the limitations discussion (e.g., why the InSAR might underestimate your subsidence) to the various parts of the methods/results/discussion where they are relevant. For example, you could say something along the lines of "InSAR may underestimate the total displacement, explaining why so and so measured more than we did, but our measurements give a good indication of blababla". Joshua Schimel ("Writing papers that get cited and proposals that get funded" calls this the "but, yes" approach, rather than "yes, but" ;-)

Conclusions:

L520: I agree with you that this assessment is probably more accurate than any of the techniques on their own, but the I still somehow stumbled over the sentence since you don't actually assess the accuracy of the different approaches against each other or against some supposed ground truth. Not a big issue, but you could consider rephrasing slightly.
L523: I must have somehow missed the part about the "most vulnerable below the current lake level", but I was wondering how you can actually assess this, if you can't actually measure displacements under the water level. If this indeed true (and I don't really doubt it, since it makes sense conceptually), it would deserve to be highlighted more prominently in the results.

**Technical corrections**
L26: insert % after ~29 (if you are using LaTeX, \SIrange{}{}{} automatically formats this correctly.
L30: *can result* instead of result
L40: impat**s**
L51: suggest removing references in this topic sentence, I don't think you have to reference a statement like "the landscape is changing"
Fig1: I suggest changing figure title (panel A does not show moraine dam). Make red things yellow in panel A (colorblind friendly)
L151: consider labeling Ombigaichen in Fig. 1

L152: tongue of the Imja Lake → remove the
L153: move "coalesced" to after "supraglacial lakes"
L219: omit two mentions of "burst"
L380: write "January through March" instead of listing the months individually.
L406: can you just write "surface geometry" instead of "surface morphometry"? (Simpler and not much different?)
L525: Note that you suddenly switched tenses (demonstrated, observed … → show, find). I suggest sticking to the past tense.
L529: Subsidence instead of "downward vertical velocity"?

---

## Author Comment (AC3)

**Reviewer 1**

The authors describe a very interesting study using a combination of InSAR and SAR feature tracking to document the degradation of the Imja lake moraine dam. Their work offers valuable insights into ongoing changes, which can be useful for hazard assessment. Generally I find the study carefully executed and the results are largely convincing. To better convey the message, however, I believe that the manuscript would benefit from some mild restructuring and occasional rewording. Below, I provide some general comments and a number of specific recommendations that I believe will address these shortcomings.

We thank the reviewer for taking the time to review our manuscript and provide helpful and detailed feedback.

**General comments**
- Given the focus of the group within which this work was performed, I am surprised that no DEM differencing was performed. Given the main findings (ice melt / surface lowering), it seems like DEM differencing would be the an important source of information? Personally, I would be delighted to see the manuscript supplemented with this information, but I recognize that this is a big ask. In any case, it would be interesting to say why this could not or was chosen not to be done.

   We now validate our results using 3D displacement measurements from very-high-resolution satellite stereo DEMs. Please see our response to Reviewer 2 for more details.

- I think the discussion falls a bit short of expectations. It nicely discusses the study results, but I think it would deserve a slightly boarder view on how and why this information matters. I have included a number of questions that the authors could consider addressing (more as a menu than an expectation that they are all addressed, and I am sure the list is not comprehensive).

   We appreciate your suggestions and have expanded the discussion to address some of your questions. Please see responses to specific comments below for details.

- The manuscript is generally well written, but frequently uses very short paragraphs, which can disrupt the reading. I recommend combining some of these in such a way that a paragraph is at least 3-5 sentences (and less than half a page). For my taste subsections in the intro are not needed (use carefully crafted topic sentences instead). A clear statement of the aim of the paper and the chosen approach to address the knowledge gap are missing in the introduction / the information is scattered throughout, consider tightening. What is true for paragraphs is also true for subsections (e.g., 4.4. and 4.5 )

   We have combined some of the shorter paragraphs to improve readability. We prefer to keep the subsections in the introduction to organize the content for quick reading. We have also worked to include a more clear statement of the problem our work seeks to address in the introduction.

A note on literature review statements: If you avoid having the citation at the start of the sentence – effectively making it the grammatical subject of the sentence – the relevant information becomes more apparent. Compare the following:

Hambrey et al. (2008) used both GPR and ERT to document buried glacier ice tens of meters thick (over 40 meters in places) overlain by up to 20 m of debris along the eastern portion of the moraine.
Vs.
Thick glacier ice – over 40 m in places and overlain with up to 20 m of debris – has been documented along the eastern portion of the moraine (Hambrey et al., 2008) [I removed the reference to GPR and ERT because you already stated this in a previous sentence]

We prefer to use citations as the grammatical subject of the sentence where it allows us to maintain active voice and clearly state who performed the work we're referring to, e.g.:
"Watanabe et al., (1995) measured the position of five painted boulders…"
Rather than:
"The position of five painted boulders was measured.."

While The Cryosphere does not offer guidelines on active vs. passive voice, many journals (e.g. Nature) suggest that authors use active voice. We reviewed the text to identify instances where this style may distract from the relevant information and modified accordingly.

**Specific comments**
L36: Consider mentioning early warning systems in addition to engineering solutions? A good reference could be Niggli, L., Allen, S., Frey, H., Huggel, C., Petrakov, D., Raimbekova, Z., Reynolds, J., Wang, W., 2024. GLOF Risk Management Experiences and Options: A Global Overview, in: Oxford Research Encyclopedia of Natural Hazard Science. Oxford University Press. https://doi.org/10.1093/acrefore/9780199389407.013.540

We referenced early warning systems in line 47. We added this reference.

L39: You jump directly to "moraine dams" here, but might want to introduce to reader first to the different types of dams that exist? A good reference to include would be Rick, B., McGrath, D., Armstrong, W., McCoy, S.W., 2022. Dam type and lake location characterize ice-marginal lake area change in Alaska and NW Canada between 1984 and 2019. The Cryosphere 16, 297–314. https://doi.org/10.5194/tc-16-297-2022

Agreed, we added a new sentence describing different ways glacial lakes can be dammed.

"Glacial lakes can be dammed by moraines, bedrock, and glacier ice (Rick et al., 2022)."

L40: Add strong precipitation events as possible trigger?

Agreed, we now include precipitation events and also ice/snowmelt events as possible GLOF triggers (Richardson and Reynolds, 2000).

"GLOFs can be triggered by rockfall, landslides, avalanches, glacier calving, precipitation and snowmelt events, and failure of the dam (Costa & Schuster, 1988; Richardson and Reynolds, 2000; Neupane et al., 2019)."

L44: It's not clear here whether you imply that the evolution of the moraine dam reduces or increases the GLOF likelihood (I guess only lowering is implied here?). Both are plausible in my mind, maybe specify?

We changed the following sentence to explicitly state that we're referring to processes that can increase GLOF likelihood, which are the focus of this study. We agree that some processes involved in moraine dam evolution may decrease GLOF likelihood (e.g. gradual consolidation of moraine material, establishment of vegetation, relaxation of steep slopes).

"As such, the evolution of moraine dams may substantially increase GLOF likelihood."

L64: It seems relatively uncommon to reference Figure 1 already in the introduction. I don't think it's necessary, and it makes the reader's life unnecessarily hard when trying to refer to the figure during the "study site" section. Consider moving it and the first reference to it to section 2.

Agreed, we moved the first instance of Figure 1 citation to Section 2.

L105: rotational failures (does it have to be rotational?) into the lake or on the other side of the dam? Either it could be a trigger for a GLOF or simply weaken the moraine. Maybe specify? You could also reference the South Lhonak Lake event.

We have revised this sentence to include all types of mass movements of moraine material (not just rotational failures) and referenced the South Lhonak Lake event. We feel that it is already clear that mass movements in general can weaken and change the shape of the moraine dam, while mass movements into the lake can result in overtopping waves.

"In addition to debris flows, other kinds of landslides have been observed on moraine dams (Richardson & Reynolds, 2000b). Sediment and ice mass movements can also indirectly contribute to GLOF risk by changing the shape and structure of moraine dams, and/or directly triggering GLOFs by creating waves that overtop the moraine dam (e.g. Zhang et al., 2025)."

L114: I think there is a conceptual error here, since feature tracking velocities are not along the line of sight.

Following Joughin et al., 2018, we perform feature tracking in radar coordinates. As stated in that article, "The horizontal along-track coordinate parallels the direction of the satellite orbit, similar to the case for optical sensors. The radar slant-range-coordinate, however, is directed in the radar line-of-sight, which is

typically at an incidence angle of 20°–45° from vertical. As a result, the slant-range offsets are sensitive to both horizontal and vertical motion."

For clarity, we edited this sentence to say: "Both InSAR and SAR feature tracking (of SLCs in original range-azimuth coordinates) can provide measurements of surface motion in the radar line-of-sight (LOS) direction."

L116: My understanding is that measuring flow velocities of glaciers is/was only possible with quite some uncertainty and only in specific locations. Maybe specify a bit more in detail when this is possible (and when not!). Saying that this can be done with mm-level precision is likely overselling it a bit.

We modified this sentence and the previous sentence, which now state:

"InSAR relates the phase offset from successive SAR acquisitions to surface displacement, potentially with mm-level accuracy (Bürgmann et al., 2000; Rosen et al., 2000). InSAR has frequently been applied to measure displacement of ice-rich features, including glaciers and ice sheets (Massonnet & Feigl, 1998; Rosen et al., 2000), rock glaciers (Bertone et al., 2022), and permafrost (Zhang et al., 2022)."

In the paragraph following these sentences, we describe limitations of InSAR that degrade measurement accuracy in practice.

L129: Consider adding reference to Jacquemart, M., Tiampo, K., 2021. Leveraging time series analysis of radar coherence and normalized difference vegetation index ratios to characterize pre-failure activity of the Mud Creek landslide, California. Natural Hazards and Earth System Sciences 21, 629– 642. https://doi.org/10.5194/nhess-21-629-2021 for the coherence.

Agreed, we have added this reference.

L153: I found myself wondering if the rate holds true for the period after the publication of the reference given here (Watanabe 2009), since almost 20 years have passed? Do you have a sense of how constant it has been over time?

Lake expansion has actually accelerated during the 1997-2020 period. We have revised this sentence to read:

"It expanded at a roughly linear rate of 0.02 km$^2$/yr until 1997, then at a rate of 0.03 km$^2$/yr from 1997 to 2020 (Watanabe et al., 2009; Gupta et al,. 2023)."

L163ff: It would be neat to plot the ERT / GPR lines of other studies (color coded by ice depth?) in Figure 1 if you can get your hands on the data. It would greatly facilitate the description of the ice (thickness) distribution. In one case you refer to "ice bottom depth" and in another place simply "40 m thick". Do these values mean the same?

We agree. We reached out to the authors of Somos-Valenzuela et al. (2012), who performed the most extensive survey (Figure R1). They shared the raw GPR data, but unfortunately the ice-bottom depth data are no longer available. In order to include the ice-bottom depths in Figure 1, we would need to reprocess and reinterpret the GPR data, which we feel is outside the scope of this effort.

These values do not mean the same thing. For clarity, we revised the beginning of this paragraph as follows:

"Geophysical studies using ground-penetrating radar (GPR) and electrical resistivity tomography (ERT) have documented the extent and thickness of buried ice in the moraine dam (Hambrey et al., 2008; Somos-Valenzuela et al., 2012; Dahal et al., 2018). Hambrey et al (2008) and Dahal et al. (2018) estimate debris thickness, but are not able to locate the bottom of the debris-covered ice, and thus present only a lower bound on ice thickness. Meanwhile Somos-Valenzuela et al. (2012) present an ice-bottom depth, but do not estimate debris thickness, resulting in only an upper bound on ice thickness."

[Figure]

**Figure R1. Ice bottom depth from Figure 8 of Somos-Valenzuela et al. (2012).**

L167: Can debris in the dam really be fully saturated while retaining its stability?

The moraine likely contains a mix of saturated and unsaturated debris (Dahal et al., 2018). In the future, we are interested in investigating the roles of saturated debris, unsaturated debris, and ice in moraine dam stability.

L174ff: I would have expected these detailed values in the discussion, rather than in the study site. I think for the description of the study site, a brief overview of where what kind of motion has been measured would suffice and be easier to remember (e.g., this corner moved most in this direction, this other part

didn't move much etc.) Decide on GNSS or GPS (Technically these should also be defined). Not clear who "the authors" are in the last sentences.

Agreed. We have updated this section to include only a general description of the motion of the moraine dam and moved the detailed values to the discussion. We checked these references, and they were GPS-only observations (not multi-GNSS observations), so we now use only "Global Positioning System (GPS)" in the text. We have also edited these sentences to make it clear we're referring to Fujita et al. (2009).

L197: consider supplementing "azimuth" and "range" with along-flight and across-flight directions such that these are defined here for future use and comprehensibility for non-radar experts.

We have added "along-track" and "across-track."

L216: consider simply saying (or specifying) that a downsampling factor of 5 was used in the range direction while the native resolution was kept for the azimuth. The "looks" terminology is very radar specific.

Agreed, "looks" is radar-specific, but we also want to be clear on the processing methodology for readers with SAR background. This sentence was revised to read:

"The insar_tops_burst workflow was used to process interferograms with five looks (effectively, a spatial averaging factor of 5) in the range direction and one look in the azimuth direction, geocoded to 20 m square pixels (Supplemental Figure 2)."

L221: how high did you chose your mean coherence for the reference and stable areas? What is the difference between the "reference" and "stable" area? Why are they different?

We selected large, contiguous areas with uniformly high coherence as near to the moraine dam as possible (Figure 1). We did not use a particular coherence threshold, but mean coherence in these areas is generally high, >0.95 year-round (Figure 2).

The reference area is used to set the reference point of each interferogram, helping to remove atmospheric noise (and potentially other spatially correlated signals that are not of interest, e.g. tectonic motion). The stable area is used primarily to assess the accuracy and precision of our displacement measurements. We have updated these sentences as follows:

"We selected a local "reference area" and "stable area" (Figure 1) as close to the moraine dam as possible with high mean InSAR coherence for the entire study period. The reference area (~0.06 km$^2$, ~144 pixels) was used to remove atmospheric noise and the stable area (~0.03 km$^2$, ~87 pixels) was used to assess measurement accuracy and precision."

L237: I presume you mean horizontal velocity here?

We're referring to the total magnitude of surface velocity here, given that feature tracking in slant range is sensitive to horizontal and vertical movement (see response to L116 above).

L245: the feature tracking is not along the line-of-sight (but along slant-range and azimuth direction).

Please see response to L116 above. We revised the L245 sentence to read:

"Since both InSAR interferograms and the component of feature-tracking motion in the slant-range direction are measurements of displacement along the radar line-of-sight, these measurements can be combined to create more accurate estimates of surface displacement over time."

L250: over what period is the median? The whole stack or the subset of pairs based on the temporal baselines?

The median is computed over the entire stack. This paragraph is intended to provide an overview of our time series inversion approach, with additional details provided in the subsequent paragraph of the original text.

L254ff: I am confused about this procedure. I am probably just misunderstanding, but here's what trips me up: First, you speak about converting the feature tracking offsets to radians. Why do you do/need this? Is it required by MintPy? Then you mask pixels with a SNR lower than 8 (how/where was the SNR calculated, and why do you choose this value?), then you remove pixels with displacement more than 10 m. But I thought you had converted the displacements to radians? So do you do the filtering first? Why do you remove measurements with a high InSAR coherence? Wouldn't the overlapping measurements provide additional robustness to your findings? I think some more details would be helpful here.

We convert our median feature tracking velocity product from pixels to radians so it is in the same units as the interferograms which serve as the other observations used to calculate the displacement time series. MintPy expects interferograms to be in units of radians.

SNR is the ratio between the peak correlation score and the mean of the surrounding correlation values. For details, we refer the reviewer to the appendix of Casu et al. (2011) and we now include this reference in the revised text. We set the SNR threshold by manually checking the resulting median velocity map for physically unrealistic noise over stable areas. A threshold smaller than 8 resulted in noisier velocity maps. A threshold larger than 8 resulted in limited improvement in noise level, while removing more measurements.

In response to a comment from Reviewer 2, we no longer use an absolute displacement threshold filter to remove displacement values greater than 10 m, so this point of confusion should no longer be an issue.

We choose not to include feature-tracking measurements where InSAR coherence is high because coherent InSAR measurements are expected to have much less noise than measurements derived from feature tracking (See response to Reviewer 2 about feature-tracking noise in the stable area). Including feature-tracking measurements in coherent areas would introduce noise without improving accuracy.

We have added additional details to the revised text that should improve the clarity of this section.

Section 4.2: You could consider adding a flow chart or other graphical representation of all the steps in your processing. Currently it's a bit hard to follow. This would also benefit from a geometric diagram, showing the LOS, the azimuth and (slant) range directions, since these can be quite confusing for non radar-experts.

Agreed. We now include a figure showing geometry information in the supplement (Figure R2) and a methods flowchart figure in the main text (Figure R3), which we believe helps to clarify the details of our processing approach.

[Figure]

**Figure R2. SAR acquisition geometry.**

[Figure]

**Figure R3. Flowchart illustrating our combined InSAR and feature-tracking approach. The time series inversion schematic is adapted from Schmidt and Bürgmann (2003). The equation relates the observed phase change (*d*) from *n* pairs of acquisitions with different temporal baselines to the change in phase between acquisitions (*m*) at a given pixel. *G* is a design matrix describing the temporal baseline of each pair, with -1 corresponding to the primary acquisition date and 1 corresponding to the secondary acquisition date. *W* is a diagonal weight matrix. *c* is the cumulative phase change time series prepared from the inversion results for each pixel.**

L268: A threshold of 0.6 is quite high, why did you chose it?

This is a threshold for the mean coherence over the moraine dam. Some parts of the moraine dam are consistently less coherent than others (Figure 2). When the mean coherence over the moraine dam is 0.6, the coherence for the fast-moving areas is low (closer to 0.4), often causing unwrapping errors (see Figure R4).

L270: I don't understand how or to what the per-pixel median feature tracking velocity products were scaled.

These products were effectively converted from a median velocity measurement to a displacement measurement by multiplying by the length of time of the entire study period. In other words, they were scaled to represent the total cumulative displacement expected from feature tracking over the full 2017 to 2024 period. This allows these values to be included in the time series inversion as a displacement measurement alongside the interferogram products.

We have edited this sentence as follows to avoid confusion: "To convert our per-pixel median feature tracking velocity (displacement rate) products into a cumulative displacement product analogous to an interferogram for the full 2017 to 2024 study period, we multiplied by the appropriate temporal baseline for the ascending (7.29 years) and descending (7.36 years) time series."

L273: What were the per-pixel displacement measurements inverted to, I don't fully understand.

We agree, this sentence is confusing. We used the interferogram observations and the scaled feature-tracking displacement product (derived from median feature tracking velocity) to invert for cumulative displacement at each time step. It is effectively the same as weighted SBAS, except the displacement product derived from feature tracking is included as an additional observation.

We have included a modified version of Figure A1 from Schmidt and Bürgmann (2003) in our new methods figure (see Figure R3 above). For this study, we append a single additional entry for the displacement product derived from feature tracking in the $G$ matrix and $d$ vector.

We have edited this sentence and added context as follows: "The cumulative feature tracking displacement product was assigned a uniform coherence of 0.6, which is the minimum allowable mean coherence for the moraine dam area. The network of per-pixel displacement measurements, including the interferograms and the ~7-year cumulative feature tracking displacement product, was then inverted using phase variance as a weight (Yunjun et al., 2019). As coherence is used to calculate phase variance, the feature tracking displacement product is weighted lower than interferograms where coherence is greater than 0.6."

Sec. 4.3: Did you do all this in MintPy or with code of your own? My understanding is that MintPy is able to do this, no?

Yes, MintPy does have a script for this, but we calculated the LOS decomposition with code of our own, which allowed us to carefully review and validate intermediate results.

Sec. 4.4: Doesn't this just belong to Sec. 4.3? I would combine ☺

We agree, and have combined these sections.

Sec. 4.5: This information seems incomplete. What did you then do with these coherence maps? Did you visually compare where there were changes and mapped these? Did you use some kind of threshold? How did you deal with snow decreasing the coherence in winter? Can you specify what months the maps cover? Is it the same as what you present in Figure 2?

We subtracted the median summer coherence values from the median winter coherence values. The result is shown in Figure 6. We interpreted the seasonal change directly, and did not use any threshold. In the absence of ground-truth data, we were wary about selecting an arbitrary threshold.

At this location, winter snow that impacts C-band InSAR coherence is uncommon (see Figure 2D+H stable area time series).

The high-coherence period extends from Jan. 1 to April 10. The low-coherence period extends from August 8 to October 7. These capture the maximum and minimum coherence, respectively and are not exactly the same as the cold-season and warm-season periods shown in Figure 2.

Sec.5.1/Figure2: - You list 8 temporal baselines, but I can only count 5, maybe 6 in the figure. Consider removing the ones that are not in the figure, if they are indeed missing?

All eight temporal baselines are included in the figure. Some are uncommon and obscured by other markers.

- I would also recommend changing the color scale of plots C,G,D, and H to something different from panels A,B,E,F because they show very different quantities, and it is initially very confusing (e.g., you could use viridis for one and magma for the other).

   Agreed, we have changed the colormap of the coherence maps.

- You outlined the area with lower coherence at the front of the lake, but the moraines along the lake also show up as low coherence areas. Why did you exclude those from the "moving area"? Is something else causing the low coherence area there? Or are they also ice-cored? (→ address in the discussion)

   We do not include these areas as part of the "moraine dam" because the surrounding ground surface is above the lake elevation. Coherence change doesn't provide evidence that these areas

are ice-cored, so hillslope processes (dry ravel, gully formation, rockfall, etc.) are more likely responsible for low coherence in these areas.

Sec.5.2/Figure3:

- I believe (as highlighted elsewhere, sorry for the repetitive comments) that the LOS you mention in connection with the feature tracking is actually the slant-range. Replace throughout.

  See response to L114 above.

- I am a little bit perplexed as to why the velocities from your feature tracking show positive velocities in both orbits. How do you interpret this? Combined with the rather low SNR over this whole area, do you believe that these are indeed horizontal displacements? Or do you think they are in fact a component of large-scale subsidence? Without getting too discussion-y, it would be nice to give a bit more context here.

  We interpret this positive signal from both orbits as positive range change corresponding to net movement of the surface away from the satellite along both LOS vectors. This positive range change can be explained in part by vertical subsidence.

- I ask this also because the same signal appears on Lhotse glacier, in the top-right corner of your panels. On that note, somewhere in the manuscript (maybe in the study section?) it would be nice to point out that there is another glacier there, which also shows a signal. That way, the readers can understand those spatial patterns.

  Similar positive signals are observed over the Lhotse Glacier, to the northwest of the Imja moraine dam. We now mention the Lhotse Glacier in the study site section (2) and results section (5.2) so readers can interpret these signals: "Immediately northwest of the moraine dam is a lobe of the debris-covered Lhotse Glacier." and:
  "Positive LOS velocities are also observed over the Lhotse Glacier northwest of the moraine dam."

Sec. 5.3/Fig.4:

- What I miss between section 5.2 and 5.3 is a comparison / context regarding the different magnitudes of motion. In 5.2 you describe the results in terms of velocities in cm/year, and then in 5.3 you provide total displacements. For me, these are hard to reconcile in my head. Can you somehow standardize?

  Section 5.2 is intended to describe the intermediate product that is included to compute the cumulative displacement time series described in Section 5.3.

  As a response to Reviewer #2 comments, we now include a section with comparative analysis of products from 1) our combined approach, 2) InSAR-only time series, and 3) SAR-feature-tracking-only time series. This new section enables direct comparison between measurements from different sources and provides additional context for sections 5.2 and 5.3.

- Between sections 5.2 and 5.3, the reader is provided with many different quantities and numbers, but it's relatively hard to do much with this information. At the end of section 5.3, you provide an overarching interpretation of the combined patterns. My suggestion would be to put that information upfront and say something like: "The observed patterns of deformation are consistent with downward and eastward motion of the moraine, as is evidenced by the feature tracking and InSAR inversion. The feature tracking shows …"
- This might even remove the requirement of having the two subsections 5.2 and 5.3.

We understand that this section provides many numbers, but they're all numbers that we feel are important to report. To make this section easier to read and keep it organized, we prefer to keep our interpretations associated with the relevant quantities, rather than frontload the interpretation and then finish with a long list of quantities.

- On a related note, I was somehow surprised not to see the results from the InSAR on its own. Can you provide these or justify in the text why you don't see the need to do so? Personally, I would very much like to see a few wrapped interferograms to judge the data quality coming from the InSAR.

As mentioned above, the InSAR-only results are now included in the main text, as detailed in the responses to Reviewer #2.  We also now include some examples of wrapped 12-day interferograms as a supplemental figure (Figure R4).

[Figure]

**Figure R4. Examples of wrapped interferograms. Note that no local reference point has been set for these interferograms, and they still contain some atmospheric noise over the moraine dam.**

- Fig.4 E-H: Is this the combined cumulative displacement or just from InSAR? The label on the yaxis should probably be cumulative displacement. Can you additionally mark (in the maps and in the profile lines) a few more points (e.g., 2 above and 2 below the mean? That would provide a bit of insight into the spatial patterns and lend credibility to your inversion.

  This is the combined cumulative displacement. We agree, and adjusted the y-axis label on the figure. We also marked the location of specific points to connect the map with the time series plots (see Figure R5).

[Figure]

**Figure R5. Cumulative displacement time series from the combined InSAR and feature tracking inversion. A–D) Maps showing total cumulative displacement of the moraine dam and stable area over the study period for the ascending (A) and descending (B) orbits, and the east/west (C) and vertical (D) directions. Marked points show the 1st, 25th, 75th, and 99th percentile pixels for cumulative ascending displacement in the moving area. E–H). Cumulative displacement time series for 50 randomly-selected pixels in the moving area (red lines) and 50 randomly-selected pixels in the stable area (grey lines). The black lines show the mean velocity of all pixels in the moving area. The other colored lines show the cumulative displacement time series for the pixels marked in the maps above.**

- Fig. 4 A-D: Personally, I would use a different colormap (e.g., magma) for the vertical displacement because you are showing a new quantity. I would also appreciate a slightly more permissive color range (+/- 3m?) to show more small scale patterns.

  We adjusted the colorbar limits to -3 to +3 m (Figure R5)

  We acknowledge that the repeated use of the RdBu colormap could be confusing. We selected it because this colormap is commonly used to show vertical displacement. It could make more sense to change the colormap for LOS movement, but we prefer to keep it, as other diverging

colormap options tend to have less color range, be less perceptually uniform, or contain colors that we find distracting.

We will be releasing all data products and notebooks used for processing, analysis and figure preparation, so others can explore custom visualizations.

Sec. 5.4:
- L380: For my taste you don't need to reiterate the velocities of the stable area here, since you clearly made the point that they are negligible.

  Agreed, we have removed these.

- Try to avoid "noun trains" (e.g. moraine dam moving area vertical velocity"). Why not say "the vertical velocity of the moraine dam increased from May to October, when it reached a maximum of …"

  Agreed, we attempted to disassemble some of the longer noun trains throughout the text.

- Rather than speaking of "vertical velocity" (here and earlier on, but here at the latest), why don't you call it subsidence? This would be more clear.

  We avoid referring to vertical velocity as subsidence because 1) subsidence only describes downward motion, and we are showing vertical motion which happens to be mostly downward (but could be upward), and 2) We want to keep our signs consistent, with negative corresponding to down. If we said "subsidence of -0.2 m" it could be misinterpreted as negative subsidence (AKA uplift).

- Fig.5: This is a detail, but I am personally bothered by the "plastered on" scale bars ;-) Can you put them next to each other (both horizontal) between panels B and C?

  Assuming you're referring to the colorbars, we prefer to leave them where they are (though we understand the issue, and will consider your suggestion during future figure preparation). There is not a lot of extra space in this figure, and as long as the colorbars are legible, we prefer to prioritize making the maps as large as possible.

- I don't believe that section 5.4 deserves its own subsection, since it's just another result of the combined inversion. I would integrate with the other part(s).

  Agreed, we have integrated this section with Section 5.3.

Sec. 5.5
- This section (2 sentences!) is a bit comical ;-). Since this comes directly from the coherence analysis, why do you not just integrate it with section 5.1? It's the obvious interpretation of your

coherence analysis, and lays the perfect justification for why you primarily detect downward motion with your other analyses.

Agreed. There was more to this paragraph/section in earlier drafts, but not in the submitted version. We integrated this "section" with Section 5.1 (InSAR coherence).

**Discussion:**

- L442: Could you compare the values for the ice flow during the cold months to what would be expected for a glacier from only the internal deformation / creep component (Cuffey and Patterson, sec. 8.3, I think)? Are they anywhere in the same range?

We actually performed this analysis and included it in earlier drafts, but removed it prior to submission due to concerns about manuscript length and the validity of the shallow-ice approximation for the Imja Lake moraine dam. We now include this analysis in the supplementary material.

The magnitude of our mean winter velocity estimate of 12.8 cm/yr is consistent with what we'd expect given the range of measured ice thickness. Based on a mean ice thickness of ~30 m (Somos-Valenzuela et al. 2012) and a mean moving area surface slope of ~10 degrees (Supplemental Figure 1), we expect a surface velocity of ~12 cm/yr (Figure R6).

"We computed the expected surface-parallel velocity ($u_s$) due to internal deformation of buried ice using a simple 1-D model with no basal sliding (Eq. R1; Eq. 8.35 in Cuffey & Paterson, 2010):

$$u_s = \frac{2A}{n+1} \tau_b{}^n H \tag{R1}$$

where $A$ is a flow rate factor (typically $2.4*10^{-24}$ $s^{-1}Pa^{-3}$ for temperate glacier ice), $n$ is the flow law exponent (3), $H$ is the ice thickness, and $\tau_b$ is the basal shear stress (Eq. R2):

$$\tau_b = \rho g H \sin(\theta) \tag{R2}$$

where $\rho$ is ice density (917 kg/m$^3$), $g$ is acceleration due to gravity (9.81 m/s$^2$), $H$ is ice thickness, and $\theta$ is surface slope. We computed expected surface-parallel velocity due to ice flow for a range of ice thicknesses (0-70 m) and surface slope values (0-20°) for the Imja Lake moraine dam."

[Figure]

**Figure R6. Expected along-slope surface velocity due to internal deformation (no sliding) of temperate glacier ice (Eq 8.35 in Cuffey & Paterson, 2010) for a range of ice thicknesses and surface slope for the Imja Lake moraine dam.**

- L468: Can you speculate about why the flow direction may have reversed? Did the overall slope change since the early 1990s?

We speculated about the change in flow direction in lines 471-479 of the original submission. We attribute the change in flow direction to 1) long-wavelength topographic inversion associated with differential ablation of buried ice (see Figure R7 and R8) and 2) changing longitudinal stress. We observe a concave-upward transverse profile (bottom right panel of Figure R7) for the Imja Lake moraine dam. A similar configuration, where the glacier surface slopes downward from the lateral and terminal moraines toward the centerline, is observed for a number of debris-covered glaciers in the Everest region (e.g., Nuptse Glacier, Figure R8). This scenario introduces driving stress toward the glacier centerline (downslope), which is consistent with observed motion in our horizontal surface velocity maps.

We hypothesize that this morphology is likely the result of differential ablation caused by some combination of spatially variable debris thickness and local topographic shading from large lateral moraines. At the Imja Lake moraine dam, the topographic inversion may be accelerated by enhanced ablation of exposed ice at the lake edge. In addition, the buried ice in the moraine dam is no longer under the compressive longitudinal stress regime experienced when previously connected to the actively flowing main glacier. Some portion of the eastward, downslope flow could be related to the evolving longitudinal stress.

To assess timescales for development of this concave-up morphology, we looked for evidence of long-wavelength slope and aspect change between the Shuttle Radar Topography Mission (SRTM) DEM acquired in February 2000 and modern WorldView DEMs (2016 and 2025). We found that the moraine dam aspect has not changed substantially since 2000.

We note that Watanabe et al. (1995) only measured movement of a few painted boulders along the outlet channel, which may not have been sufficient to characterize the broader displacement patterns at the time. Given limited observational data, we are unable to constrain the timing of the transition from mostly westward to mostly eastward moraine dam surface movement, and this is beyond the scope of the current manuscript.

We now include additional discussion on these topics in Section 6.2 of the manuscript, and include Figures R7 and R8 as supplemental figures.

[Figure]

**Figure R7. Schematic showing topographic inversion at the Imja Lake moraine dam. Melt is enhanced at the lake edges through interaction with liquid water and ice cliff retreat. Melt is suppressed near the lateral moraines due to insulation from thicker debris and the effects of local shading.**

[Figure]

**Figure R8. The Lower Nuptse Glacier, located to the west-northwest of Imja Lake. Note the general surface aspect (B), concave-up transverse profile (D) and pattern of surface motion toward the glacier centerline (C/E).**

- L471: Some confusion here: are you saying that the direction of the ice flow mentioned above (L466) was actually the direction of the ice flow of the original glacier? Or were these measured post separation?

  Yes, we assume that the original ice flow direction was westward. These measurements were made after the formation of the lake.

- L475: Are you saying here that Imja Lake is actually still a supraglacial lake? Or what do you mean by "complete decoupling from the active glacier will remove compressive stresses"? Specify clearly (and here in the Study Site, for I certainly missed this if it is at all addressed there).

  No, while there may be remnant glacier ice in the lateral moraines and/or beneath the lake, the moraine dam is likely completely decoupled from the main glacier. Here we are referring to historic changes as the lake expanded. The paragraph now reads:

  "The observed change in flow direction is likely related to evolving thickness of lower Imja Glacier though differential ablation both above and below the lake surface (Kjær & Krüger, 2001), changing longitudinal stresses associated with decoupling from the active flowing glacier during retreat, and interaction with the expanding lake. Moraine dam topography and lake

bathymetry suggests that buried ice within the moraine dam is likely thicker than ice under the lake to the east, resulting in a west to east ice thickness gradient (Haritashya et al., 2018). Complex interactions between the evolving glacier and the growing lake to the east would likely also recharacterize the stress regime, as has been observed for lake-terminating glaciers in the region (King et al., 2018). Our detailed displacement time series can be used to constrain ice and debris evolution models, which are needed to better understand the Imja Lake moraine dam evolution and implications for changing structural stability."

- Sec. 6.3: As elsewhere, I would integrate these findings with the discussion of the subsidence.

Agreed, we have integrated this section into Section 6.1 (Ice melt).

Questions that I would like / was expecting to see addressed in the discussion:

We have added additional text to the discussion to address most of these questions.

- How does the subsidence / ice melt compare to measurements from other places? Can you estimate from this how long you expect the ice core to stick around? What will Imja lake look like, once it's all gone? How does it affect the hazard?

Downwasting rates vary widely and depend on climatology and debris thickness, but our vertical velocity measurements are in the same range as previous estimates (Table R1). We have added this table to the supplemental materials.

We initially included a section extrapolating the observed velocity to calculate the number of years until the surface is at the current lake level. However, we ultimately felt that this exercise required too many assumptions about future velocity and lake levels, so we chose to remove it.

When the buried ice in the moraine dam is gone, Imja lake will likely be dammed only by the terminal moraine, similar to Rolpa Lake. This will increase GLOF risk, as the moraine dam will be narrower and weaker, and the lake volume will be larger.

**Table R1. Summary of published downwasting rates for ice-cored moraines. Expanded from Irvine-Fynn et al. (2011).**

| Downwasting rate (m/yr) | Location | Reference |
|---|---|---|
| <4.8 | Jotunheimen, Norway and Kebnekaise, Sweden | Østrem (1965) |
| 0.048-0.08 | Southeastern Alaska, USA | Mckenzie (1969) |
| 0.1 | Yukon, Canada | Ross (1976) |
| 0.003-0.3 | Yukon, Canada | Driscoll (1980) |
| <5 | Alberta, Canada | Mattson et al. (1991) |
| 1.0 | Yukon, Canada | Johnson (1992) |
| 0.4-4.8 | Punjab Himalaya, Pakistan | Mattson (1993) |
| 0.05 | Svalbard | King & Volk (1994) |
| 0.1-0.4 | Svalbard | Etzelmüller (2000) |
| 0.3-2.5 | Kötlujökull, Iceland | Krüger & Kjær (2000) |
| 0.4-3.5 | Svalbard | Lukas et al. (2005) |
| 0.9 | Svalbard | Schomacker & Kjær (2008) |
| 0.7 | Svalbard | Irvine-Fynn et al. (2011) |
| 0.16 | Svalbard | Tonkin et al. (2016) |
| 0.5 | South Tyrol, Italy | Kunz et al. (2021) |

- What implications do you think these kinds of measurements can have for operational monitoring. What drawbacks or caveats?
- Under what circumstances do you think your approach can work (in terms of geometry of the moraine/valley)? Would this work on most glacial lakes in HMA or is there something special about this one (on that note, a justification about why you chose Imja in the Study Site description would be nice).

Here we address both reviewer questions together. In light of the reviewer's later comment about our limitations section, we have replaced that section with a new discussion section called "Advantages and limitations for operational monitoring." This section contains some text from the original limitations section, along with text addressing the specific reviewer questions raised above.

These kinds of measurements can be integrated into hazard assessments to guide future allocation of operational monitoring resources. For example, glacial lakes that are found to be experiencing rapid moraine degradation could be surveyed more frequently than lakes where the moraine dam is not changing. We are preparing a follow-on manuscript scaling the methodology and analysis presented here to dozens of other lakes in the region, and plan to discuss these topics in more depth there.

- All the numbers presented in the Study Site (with regards to the changes measured by others) should be presented in the discussion, I think, giving context and comparability of your results. If possible, it would be really cool to put all the different numbers on map.

Agreed, we have moved most of these numbers to the discussion.

S6.4: It unfortunate to end your strong manuscript with a long discussion of the limitations, because you leave the reader with information that suddenly calls into question all the work you just presented. As a reader, it requires me to re-evaluate what I just read and put the interpretations in light of the limitations I am only provided with now. I would recommend moving the different parts of the limitations discussion (e.g., why the InSAR might underestimate your subsidence) to the various parts of the methods/results/discussion where they are relevant. For example, you could say something along the lines of "InSAR may underestimate the total displacement, explaining why so and so measured more than we did, but our measurements give a good indication of blababla". Joshua Schimel ("Writing papers that get cited and proposals that get funded" calls this the "but, yes" approach, rather than "yes, but" ;-)

Agreed, we have moved much of this discussion to related sections. As described above, we have replaced the existing limitations section with a section that described advantages and limitations of our approach for operational monitoring.

**Conclusions:**
- L520: I agree with you that this assessment is probably more accurate than any of the techniques on their own, but the I still somehow stumbled over the sentence since you don't actually assess the accuracy of the different approaches against each other or against some supposed ground truth. Not a big issue, but you could consider rephrasing slightly.

  Agreed. We now validate our results using 3D displacement measurements from very-high-resolution satellite stereo DEMs, which offers additional support for this statement.

- L523: I must have somehow missed the part about the "most vulnerable below the current lake level", but I was wondering how you can actually assess this, if you can't actually measure displacements under the water level. If this indeed true (and I don't really doubt it, since it makes sense conceptually), it would deserve to be highlighted more prominently in the results.

  See lines 435-438: "If we extrapolate the observed 2017 to 2024 mean subsidence rates forward in time, and consider the current moraine dam topography, we find that the northeast area of the moraine dam bordering the lake edge is most vulnerable to subsidence below the current lake level. This topographic evolution would result in westward expansion of Imja Lake, decreasing the width of the moraine dam."

**Technical corrections**
- L26: insert % after ~29 (if you are using LaTeX, \SIrange{}{}{} automatically formats this correctly.
- L30: can result instead of result
- L40: impats
- L51: suggest removing references in this topic sentence, I don't think you have to reference a statement like "the landspace is changing"
- Fig1: I suggest changing figure title (panel A does not show moraine dam). Make red things yellow in panel A (colorblind friendly)
- L151: consider labeling Ombigaichen in Fig. 1

- L152: tongue of the Imja Lake → remove the
- L153: move "coalesced" to after "supraglacial lakes"
- L219: omit two mentions of "burst"
- L380: write "January through March" instead of listing the months individually.
- L406: can you just write "surface geometry" instead of "surface morphometry"? (Simpler and not much different?)
- L525: Note that you suddenly switched tenses (demonstrated, observed … → show, find). I suggest sticking to the past tense.
- L529: Subsidence instead of "downward vertical velocity"?

We addressed all of these issues in the revised text, thank you!

**Reviewer 2**

Dear Editor, Dear Authors,

Thank you for the opportunity to review your manuscript. I appreciate the effort and dedication that has gone into developing your study and the processing strategies employed. However, after a thorough evaluation, I regret to inform you that I must recommend rejecting this paper in its current form.

While the work demonstrates potential, it would benefit significantly from the involvement of an InSAR/SAR expert with specific expertise in time series analysis development. Such expertise could provide critical support to strengthen the analysis and ensure the robustness of the methodologies employed.

We thank the reviewer for taking the time to review our manuscript and provide feedback.

We note that all authors have experience with InSAR time series processing (e.g. Henderson et al., 2013; Allstadt et al., 2015; Henderson et al., 2017; Brencher et al., 2021; Brencher et al., 2025).

Additionally, the reliance on open-access code, while commendable for accessibility, may introduce fundamental flaws in the processing approach if not carefully validated.

We respectfully disagree with this opinion. As detailed in our methodology, we are using ISCE2 and MintPy for the majority of our processing workflow, both open-source software packages with extensive community validation, operational use, and a demonstrated record of robustness among the scientific community. Using reproducible, open-source tools and open data also enables others in the community to independently assess our methods in the future. We now include Fig. R3 in our methods section to improve clarity on our approach.

Below, I outline the key reasons for this recommendation, along with specific feedback.

**Lack of Acknowledgment of Prior Work:** Your study builds upon well-established methods, such as those introduced by Casu et al. (2011) and Jolivet and Simons (2018), yet these foundational works are neither appropriately acknowledged nor referenced. This omission undermines the context of your research and does not allow the reader to contextualize your research in literature.

To better contextualize our work, we expanded our literature review, and now include a brief introductory paragraph on InSAR and feature-tracking time-series methods. We now summarize the methods outlined by Casu et. al. (2011) and Jolivet and Simons (2018) and cite them in the main text. We also included additional text clearly stating how our methods differ from these studies.

**Introduction of Non-Validated Processing Approaches:** The manuscript introduces non-standard processing techniques, such as the use of pixel offsets to constrain an InSAR time series, without sufficient validation or precedent in the literature. Introducing novel methodologies is commendable;

however, without proper validation, these methods cannot be reliably assessed or accepted by the broader scientific community.

To our knowledge, pixel offsets have not been previously used to constrain an InSAR time series. However, many previous efforts outline methods to prepare displacement products by combining measurements from pixel offsets and InSAR (e.g. Joughin, 2002; Gray et al., 2005; Liu et al., 2007; Sánchez-Gámez and Navarro, 2017; Joughin et al., 2018), which were all cited in the original submission.

We agree that validation using independent measurements would be ideal to assess the performance of our approach. Unfortunately, we do not currently have access to any in-situ GNSS or other validation data acquired during our study period for these ice-cored moraines. We hope to coordinate future field campaigns to acquire additional validation data for these features.

To respond to comments from both reviewers requesting additional validation, we prepared precise 3D displacement measurements (with uncertainty estimates) from very-high-resolution satellite stereo DEMs, and used them to validate our combined InSAR and feature tracking displacement time series. We find that measurements made using our time series approach agree with these DEM-derived measurements. We also show that our method offers an improvement over traditional InSAR-only SBAS or SAR-feature-tracking-only SBAS.

We now include the following new section containing Figure R13 and tables R2 and R3 in the main text, with detailed processing information and additional Figures R9-R12 in the supplement.

**Preparation of a high-resolution 3D displacement validation dataset from stereo DEMs**

We identified and processed two Maxar WorldView-3 in-track stereo image pairs spanning the 2017-2024 study period, including the February 11, 2016 WorldView-3 stereo pair (catalog IDs 104001001854B000, 10400100175C2D00) used by Haritashya et al. (2018), and a new in-track stereo pair acquired on January 30, 2025 (10400100A0D45D00, 10400100A193EE00). We used the latest version (3.5.0-alpha) of the Ames Stereo Pipeline (Shean et al., 2016; Beyer et al., 2018) and the processing settings outlined by Bhushan and Shean (2021) and Bhushan et al. (2024) to prepare the DEMs with 1 m posting.

We co-registered the January 30, 2025 DEM to the February 11, 2016 DEM using the demcoreg (Shean et al., 2023) implementation of the Nuth and Kääb (2011) algorithm (Fig. R9). We then corrected residual in-track "jitter" and cross-track "CCD array geometry" artifacts (see Shean et al., 2016) by fitting a Savitzy-Golay filter (window of 101 px, 2nd-order polynomial basis function) to median of elevation difference residuals over static surfaces for each row and column (Fig. R10; see ct_at_correction_wrapper function of demcoreg/coreglib.py). We computed final cumulative elevation change and elevation change rate maps for the ~9-year period (Fig. R11). The final elevation difference residuals over all exposed static control surfaces used during co-registration had a median of 0.00 m and normalized median absolute deviation of 0.14 m (<0.5 px) for the ~9-year period, corresponding to a 1-sigma uncertainty of ~1.6 cm/yr. The elevation difference residuals over the stable area used for calibration of the InSAR and

feature-tracking results (Figure 1) had a mean bias of -0.54 cm and a standard deviation of 7.77 cm, corresponding to apparent vertical displacement rates of -0.06 cm/yr and 0.87 cm/yr, respectively.

[Figure]

**Figure R9: Co-registration results for the February 11, 2016 DEM (reference) and January 30, 2025 DEM (source). The latter was shifted (+0.94 m, -2.01 m, +0.24 m) to minimize residuals over the unmasked surfaces (white) in the top right figure. Bottom row shows elevation difference maps before co-registration (left), and after co-registration (center). Bottom right map shows enhanced color stretch and histogram shows unmasked values before and after co-registration. Note that some large negative values (-5 to -2.5 m) are observed for unmasked pixels near glacier margins are included in the histogram, but these outliers did not affect the robust co-registration.**

[Figure]

**Figure R10: Row (left panel) and column (right panel) median elevation difference values (black) over static control surfaces after co-registration (see Figure R9), showing residual artifacts due unmodeled attitude error ("jitter", left) and detector array geometry calibration (right). See Shean et al. (2016) for more details. Red line shows the smoothed model from a Savitzky-Golay filter, which was used to correct all rows and columns in the unmasked elevation difference product.**

[Figure]

**Figure R11: Elevation difference maps (top row) and histograms (bottom row) of residuals over exposed surfaces (white in inset axes) assumed to have no elevation change during the study period. Left shows original difference map, center shows difference map after co-registration, right shows difference map after co-registration and correction of row/column median values.**

Following co-registration, we created shaded relief maps for the two DEMs using the "combined" hillshade option in the gdaldem utility (Roualt et al., 2024). We digitized the lake and pond shoreline using the 2025-01-30 orthoimages and shaded relief map, and masked the hillshade products over water. We then used the Ames Stereo Pipeline correlator (disp_mgm_corr.py utility of Bhushan and Shean [2025]) to prepare dense horizontal sub-pixel displacement products to track surface motion in the hillshade products, following the methods outlined in Bhushan et al. (2024). The final horizontal velocity (m/yr) components (Fig. R12) were calculated based on the 1 m DEM pixel size and the ~9 year time period. We computed the mean and standard deviation of the east/west (-1.5 cm/yr, 2.3 cm/yr) and north/south (-2.0 cm/yr, 1.9 cm/yr) velocity components over the stable area, and estimated the uncertainty of these horizontal velocity components over the Imja Lake moraine dam based on the RMSE of 2.75 cm/yr.

[Figure]

**Figure R12: Long-term (~9-year) east/west, north/south, and vertical surface velocity calculated from feature tracking of co-registered 1-m DEM hillshade products and a vertical DEM difference map. These products document the spatial extent and magnitude of moraine dam surface motion during the study period with enough detail to capture signals associated with ice cliff retreat, downslope flow, and cut/fill and downstream sediment deposition associated with the October 2016 lowering project (UNDP, 2012; Khadra, 2016). These products serve as validation for the combined InSAR and SAR feature-tracking time series, which offer detailed temporal evolution of these processes (Figures 4-5), with reduced spatial resolution.**

**Validation of combined InSAR and SAR feature-tracking products**

To enable direct comparison of our time series results with the DEM-derived validation data, we first masked pixels with a cold-season coherence of <0.7 (Figure 2). These areas contain rapid winter displacement caused by backwasting of ice cliffs, which have large signals in the DEM difference products, but we do not expect to measure with Sentinel-1 SAR observations (Figure R1). We smoothed the 1-m DEM-derived products to match the SAR feature tracking resolution using a gaussian kernel size of 7 by 7 px (~140x140 m), and sampled at the 20 m grid of velocity products for direct intercomparison. We computed zonal statistics (mean, standard deviation) for each velocity component (up/down, east/west) over both the moving and stable areas (Table R2, R3). We also computed $R^2$ values of the per-pixel differences between the InSAR/SAR velocity and the corresponding DEM validation component.

**Comparison with InSAR-only and SAR-feature-tracking-only SBAS time series**

We prepared two additional time series from the same Sentinel-1 SLC products: a standard InSAR-only SBAS time series and a SAR-feature-tracking-only SBAS time series. The InSAR SBAS time series used the same processing parameters as described for our combined approach, but does not include the scaled median velocity product from feature tracking for the inversion (blue row in Figure R3). The SAR-feature-tracking-only SBAS time series included the same set of 4, 5, and 6-year temporal baseline pairs from our combined approach (192 ascending and 210 descending), as well as all possible image pairs with a temporal baseline of three years (182 ascending and 132 descending), to ensure that the system was overdetermined. The SAR feature-tracking-only SBAS time series inversion used all available (ascending n=374, descending n=342) feature tracking displacement offsets in the slant-range direction.

The vertical and east-west velocity maps from each time series approach are shown in Fig. R13. Summary statistics for each approach and the DEM validation dataset are presented for the moving area in Table R2

and the stable area in Table R3. For the SAR-feature-tracking-only velocity products, we filtered outliers over the stable area using a 3-sigma threshold before calculating the mean and standard deviation.

[Figure]

**Figure R13. Mean vertical (top) and east/west (bottom) velocity products from InSAR-only SBAS (left), feature-tracking-only SBAS (left center), our combined InSAR and SAR feature tracking approach (right center), and downsampled DEM-derived validation data (right). Our combined approach provides the best agreement with the spatial distribution and magnitude of the downsampled DEM-derived rates.**

Our analysis shows that both the magnitude and the direction of the combined InSAR and feature tracking time series approach are consistent with the corresponding DEM validation components. The mean vertical displacement in the moving area from DEM differencing was -15.0 cm/yr. Mean vertical displacement from our combined InSAR and feature-tracking time series approach was -13.0 cm/yr. The InSAR-only SBAS underestimates overall velocity. The feature-tracking-only SBAS is able to more accurately capture long-term rate magnitude, but is unable to capture short-term variability and contains large errors.

**Table R2. Moving area velocity from InSAR-only SBAS, feature-tracking-only SBAS, and our combined InSAR and feature tracking SBAS approach compared to moving area velocity from the DEM validation data. The $R^2$ values were computed from per-pixel differences of each approach with the corresponding DEM validation data.**

| Moving area (cm/yr) | Mean (U/D) | Mean (E/W) | Std (U/D) | Std (E/W) | $R^2$ (U/D) | $R^2$ (E/W) |
|---|---|---|---|---|---|---|
| InSAR-only SBAS | -6.6 | 3.9 | 3.5 | 5.1 | 0.29 | 0.71 |
| Feature-tracking-only SBAS | -17.4 | 1.2 | 11.3 | 7.8 | 0.68 | 0.50 |
| Combined SBAS | -13.0 | 5.0 | 11.4 | 5.6 | 0.77 | 0.46 |
| DEM-derived validation | -15.0 | 8.1 | 10.1 | 8.7 | – | – |

**Table R3. Stable area velocity from InSAR-only SBAS, feature-tracking-only SBAS, and our combined InSAR and feature tracking SBAS approach compared to stable area velocity from the DEM validation data.**

| Stable area (cm/yr) | Mean (U/D) | Mean (E/W) | Std (U/D) | Std (E/W) |
|---|---|---|---|---|
| InSAR-only SBAS | 0.1 | 0.2 | 0.1 | 0.1 |
| Feature-tracking-only SBAS | -4.0 | -6.2 | 4.0 | 5.2 |
| Combined SBAS | 0.0 | 0.0 | 0.1 | 0.1 |
| DEM-derived validation | 0.1 | -1.1 | 0.1 | 0.7 |

There are some small (~2-3 cm/yr) differences in the magnitude of the components from our combined InSAR and SAR feature tracking results and the DEM validation data. Some disagreement is expected here based on limitations of the InSAR and feature-tracking methods, which we discuss at length in the Introduction and Discussion section of the manuscript. We interpret the data with these limitations in mind, and the conclusions of our manuscript have not changed. In summary, our combined time series approach offers the best agreement with the DEM validation data, especially for the up/down motion, which is likely more relevant for GLOF hazard.

**Integration of Processing and Physical Interpretation:** Attempting to combine an alternative processing strategy with the interpretation of a physical phenomenon adds complexity to the manuscript. The two aspects require independent development, validation, and presentation to ensure clarity and scientific soundness.

Given these concerns, I suggest splitting the manuscript into two separate papers:
- A technical paper presenting and validating the alternative processing approach, potentially submitted to a journal specializing in remote sensing or geophysical methods (e.g., IEEE Transactions on Geoscience and Remote Sensing).
- A separate study presenting the physical results and interpretation, submitted to a journal like The Cryosphere.

We appreciate the reviewer's suggestion for an alternative publication strategy, but we do not agree that it is necessary. The unique physical processes driving the observed change of the Imja moraine dam involve spatial and temporal variability that cannot be reproduced at a commonly used InSAR calibration site. This limits the ability of a standalone technical paper to demonstrate the effectiveness of our approach for the relevant signals observed at Imja Lake. Furthermore, a standalone scientific paper without detailed methodological considerations (and associated validation) for this specific site could leave readers without the appropriate context to interpret our results.

We are confident that the new DEM validation and alternate SBAS time series inversion comparisons address the reviewer's main criticism. These efforts offer increased confidence in our approach, and

support our interpretation of the underlying physical phenomena. Furthermore, our reproducible methods, open-source code and open datasets enable others in the community to further evaluate our results for the Imja Lake moraine dam and other sites with suitable Sentinel-1 coverage.

We feel that a single, combined manuscript in The Cryosphere is the best option for this work, and that it will appeal to a broad audience.

**Major Concerns and Specific Feedback:**
**Atmospheric Noise Filtering (Lines 202-228):**
The methodology for removing atmospheric noise and assessing measurement accuracy is not specified. This step is critical for validating the results.

For clarity, we revised the relevant text in the methods section, which now reads:
'We selected a local "reference area" and "stable area" (Figure 1) as close to the moraine dam as possible with high mean InSAR coherence for the entire study period. The reference area (~0.06 km$^2$, ~144 pixels) was used to remove atmospheric noise and the stable area (~0.03 km$^2$, ~87 pixels) was used to assess measurement accuracy.'

We also added additional text to our methods section:
"Following the time series inversion, we computed and removed the median apparent displacement over the reference area from the LOS displacement estimate at each time step to mitigate atmospheric noise. The standard deviation of the median apparent displacement values over the reference area was 3.1 cm for the ascending data (n=214) and 0.5 cm for the descending data (n=227). The moraine dam is approximately 900 by 800 m and the reference area is only 250 m northwest of the moraine dam. Given the expected correlation length scales of several km for atmospheric noise in mountains (e.g. Bekaert et al., 2015), the apparent displacement in the reference area should be similar to the atmospheric noise over the moraine dam, and we subtract the reference area median displacement from each interferogram. To evaluate remaining measurement uncertainty, we first computed the mean apparent displacement in the stable area at each time step, then computed the standard deviation of this mean over all time steps to characterize the magnitude of remaining atmospheric noise at a given time step. The standard deviation was 0.2 cm for the ascending orbit and 0.4 cm for the descending orbit"

**Kernel and Window Sizes (Lines 228-231):**
The chosen dimensions and skip sizes are not sufficiently justified. It would be helpful to provide a sensitivity analysis or rationale based on the study's objectives.

We agree that additional justification for these decisions would improve our manuscript. We added additional text to the methods section, which now reads:
"We coregistered all SLCs and performed feature tracking on the SLC products in radar coordinates at the native resolution using a modified version of the insar_tops_burst workflow and the 'dense ampcor' ISCE2 feature tracking routine. We selected kernel sizes and skip sizes based on the characteristics of the moraine dam and an empirical sensitivity analysis.

The moraine dam is roughly square (~800 by 900 m) in ground range. To maximize the number of unique kernels over the moraine dam that do not extend beyond the edge of the feature, we chose to use kernel sizes and skip sizes that were roughly square in ground range. Kernels containing pixels over both the moraine dam and the surrounding area are likely to produce poor matches due to intra-kernel variability in the magnitude and direction of surface displacement.

Smaller kernel sizes will result in poor matching, while larger kernel sizes will reduce the spatial resolution of the output feature tracking measurements. We found that a kernel size of 10 pixels in azimuth and 60 pixels in range, (141 by 138 m in ground range) was the smallest possible square kernel size that provided high-quality matches. We used a skip size of 4 pixels in azimuth and 20 pixels in range, or 56 by 46 m in ground range, to minimize interpolation without increasing processing time. Smaller skip sizes substantially increased processing time with negligible improvement, while larger skip sizes required more interpolation between sparse measurements. The search window size was set to 10 pixels in azimuth and 30 pixels in range, or 141 m by 69 m, much larger than the expected cumulative surface displacement of the moraine dam over the study period. Following computation of pixel offset values using the above parameters, we used bilinear interpolation to fill residual gaps to ensure spatial continuity for the inversion."

**Displacement Accuracy (Lines 235):**
The theoretical accuracy of feature tracking measurements should account for terrain slope and ground range, not just slant range, as this directly impacts the reliability of your results.

We agree and added additional text considering the terrain slope and ground range to the methods section:

"To calculate the theoretical accuracy of our feature tracking measurements, we first calculated the ground-range resolution of our slant-range SLCs. Following Notti et al., (2010) we computed the ratio of slant-range resolution to ground-range resolution (R-index) for our SLCs, considering terrain slope and aspect and the radar LOS vectors. We use this index to calculate the resolution of our SLCs in the ground-range direction. Over the moraine dam, the median ground-range resolution was 4.38 m for the ascending SLCs and 3.82 m for the descending SLCs. Assuming perfect SLC coregistration and a conservative feature-tracking precision of 0.2 pixels (Fialko & Simons, 2001; Strozzi et al., 2002), the theoretical accuracy of our feature tracking measurements is 2.82 m in azimuth, 0.88 m in range for the ascending SLCs, and 0.76 m in range for the descending SLCs. Given the expected mean surface velocity of <1 m per year for the moraine dam (Fujita et al., 2009), we generated feature tracking offsets with multi-year temporal baselines to maximize displacement between acquisitions. As longer temporal baselines could result in degraded quality due to significant changes in surface scatterer characteristics, we generated offsets for all possible feature tracking image pairs with temporal baselines of 4, 5, and 6 years (Table 1). In total, we generated 192 ascending and 210 descending feature tracking offset maps with corresponding signal-to-noise ratio (SNR) maps. Our expected precision in the range direction is 0.22 m/yr, 0.18 m/yr, and 0.15 m/yr for our 4, 5, and 6-year feature tracking offset maps, respectively."

We also added the following figure to the supplement (Fig. R14):

[Figure]

**Figure R14. Ground-range resolution of ascending (A) and descending (B) SLCs. C and D show the distribution of ground-range resolution for pixels over the moraine dam (white outline).**

**Feature Tracking and InSAR Constraining (Lines 250-260):**
Using pixel offsets to constrain a high-accuracy phase time series is methodologically questionable.

Where InSAR phase is coherent, it offers much better accuracy than feature tracking offsets. However, during the months of August and September, when coherence is low over the moraine dam moving area in the 12-day interferograms (Figure 2), the unwrapped phase alone does not accurately capture much of the moraine dam surface movement. The phase time series is not "high-accuracy" during this part of the year, which leads to underestimation of the long-term InSAR-only rates (see comparison in earlier response).

By including the scaled median pixel offset velocity product in our time series inversion, we provide additional information that constrains the time series solution during periods when the phase information is not accurate.

Errors inherent to pixel offsets should be filtered using metrics like SNR rather than aggregating them into a median velocity product.

We agree that relying on individual feature tracking products rather than an aggregated median velocity product could theoretically have advantages for characterizing time-dependent behavior.

We do filter our feature tracking pixel offsets using SNR prior to aggregating them into a median velocity product. However, examining feature tracking velocity product values in the range direction over the stable area demonstrates that even feature tracking measurements with high SNR (>8) contain 1-sigma random error of ~3-4 cm/yr (Fig. R15). By computing the median velocity across many feature tracking products, we mitigate this random error. The spatial mean of per-pixel median LOS velocity over the stable area was 0.7 cm/yr for the ascending track and -0.3 cm/yr for the descending track. The resulting low-noise median velocity product can then be applied to constrain the InSAR time series during combined inversion as described above.

[Figure]

**Figure R15. Apparent stable area feature tracking velocity values in the slant-range direction for ascending and descending products for all pixels with SNR >8. Note spread of values, despite high SNR. We assume that the standard deviation provides an estimate for the random 1-sigma error of feature tracking velocity in the slant-range direction.**

**Displacement Thresholding (Line 256):**

Applying a displacement threshold when the expected values are already known introduces bias into the analysis. This approach is contradictory to the principle of deriving unbiased results.

Agreed, we removed the displacement threshold from our revised analysis and updated the text accordingly.

**Methodology Validation (Lines 261-265):**

Using the normalized median absolute deviation (NMAD) for variability assessment is unconventional for SAR analysis. A focused validation or comparative analysis with established methods is required.

We do not agree with the reviewer here. Just because a statistic may appear "unconventional" to a specific community does not mean that it cannot provide useful information. Nevertheless, we changed this panel to show more "conventional" standard deviation rather than NMAD values.

See earlier response for validation and comparative analysis with established methods.

**Continuous Network Requirement (Lines 268):**
Removing interferograms with low coherence to maintain a continuous network is unnecessary. The NSBAS approach (Jolivet and Simons, 2018) allows for network separations, negating the need for such constraints.

For this study, we focused on methods implemented in actively-maintained (Python 3.X support) open-source InSAR processing software, specifically MintPy, which does not currently include the NSBAS approach. While we are certainly interested in testing an NSBAS approach in the future, we feel that re-implementing this is beyond the scope of this manuscript, and that the new validation and comparison with other SBAS methods sufficiently justifies our approach.

**Pseudo 3D Analysis (Figure 4):**
Ignoring the north component in your analysis leads to incomplete results, especially in complex topography. Conducting a full 3D inversion using all four available directions would yield more accurate and meaningful results.

We first note that only a single ascending and a single descending Sentinel-1 track cover this site. We assume by "all four available directions" the reviewer is referring to an inversion using feature-tracking offsets in the slant-range and azimuth direction for the ascending and descending orbits (e.g. Samsonov et al., 2021). However, this approach suffers from the same issue as the feature-tracking-only SBAS time series we prepared for the earlier comparison. Specifically, individual feature tracking products are too noisy to effectively capture the seasonal variability of moraine dam velocity. Feature tracking offsets in the azimuth direction are additionally expected to have less precision than feature tracking offsets in range due to coarser azimuth resolution, substantially increasing the magnitude of the noise. For example, Fig. R16 shows the apparent feature tracking velocity in the azimuth direction over the stable area, with random 1-sigma error of 16.3 cm/yr (ascending) and 24.4 cm/yr (descending), much larger than the corresponding random error for slant-range displacements in Fig. R15. Given this noise, a full 3D inversion that uses feature-tracking offsets in the azimuth direction is not viable for the Imja Lake moraine dam. We address concerns regarding north/south motion in subsequent responses.

[Figure]

**Figure R16. Apparent stable area feature tracking velocity values in the azimuth direction for ascending and descending products for all pixels with SNR >8. Note the spread of values. We assume that the standard deviation provides an estimate for the random 1-sigma error of feature tracking velocity in the azimuth direction.**

**Topographical Signal Contamination (Figure 5, Panel B):**
The downward signal observed in July-October suggests north-south displacement contamination. A proper 3D analysis would mitigate this issue and provide clearer insights.

To investigate this issue, we used the observed "true" north/south velocity measurements from the DEM feature tracking analysis (see Figure R12) to estimate the expected bias in the east/west and vertical velocity components from our decomposition. To do this, we solved the following equations (Eq. R3, Eq. R4) for vertical and east/west velocity:

$$v_{asc} = \hat{l}_{asc} \cdot \hat{l}_{ud} * v_{ud} + \hat{l}_{asc} \cdot \hat{l}_{ew} * v_{ew} + \hat{l}_{asc} \cdot \hat{l}_{ns} * v_{ns} \tag{R3}$$

$$v_{des} = \hat{l}_{des} \cdot \hat{l}_{ud} * v_{ud} + \hat{l}_{des} \cdot \hat{l}_{ew} * v_{ew} + \hat{l}_{asc} \cdot \hat{l}_{ns} * v_{ns} \tag{R4}$$

Where $\hat{l}$ represents a unit vector, $v$ is the mean velocity over the study period, and the subscripts ($asc$, $des$, $ud$, $ew$, and $ns$) correspond to the ascending, descending, up/down, east/west, and north/south components, respectively. The $v_{asc}$ and $v_{des}$ values are the mean ascending and descending LOS velocity magnitude from our combined InSAR feature tracking time series approach. For this analysis we substitute the "true" north/south velocity from the DEM validation analysis as the $v_{ns}$ value in each equation. Thus, rather than assuming that the $v_{ns}$ north/south contribution to the LOS velocity is 0, as we

do in the manuscript, we include the "true" north/south velocity magnitude when solving for the $v_{ud}$ and $v_{ew}$ components (Figure R18).

[Figure]

**Figure R18. Estimated bias in up/down and east/west velocity components caused by ignoring the north/south velocity in the LOS decomposition.** *v* is the mean velocity over the study period, and the subscripts (*ud*, *ew*, and *ns*) correspond to the up/down, east/west, and north/south components, respectively. **(left) The north/south velocity components from the DEM validation dataset. (One from left) the up/down and east/west velocity components, assuming that the north/south velocity component is 0. (Center) The up/down and east/west velocity components calculated using the north/south velocity component from the DEM validation dataset. (One from right, right) Estimated bias caused by neglecting the north/south velocity component.**

The DEM validation data show that the northern portion of the moraine dam is moving southward, with a mean velocity of -19 cm/yr, while the southern portion of the moraine dam was moving northward, with a mean velocity of 7 cm/yr. Including these true displacements during the LOS decomposition leads to a slight underestimation of the vertical velocity magnitude over the northern portion (mean bias of 1.6 cm/yr or ~10%) and a slight overestimation of the vertical velocity magnitude over the southern portion (mean bias of -2.1 cm/yr or ~35%). When averaged over the entire moving area, these biases largely cancel, with a mean total bias of +0.6 cm/yr, or ~4% of the observed mean vertical velocity from the DEM validation data. In the east/west direction, including the true north/south displacements results in a slight overestimation of eastward velocity (mean bias of 0.2 cm or 3% of the observed mean east/west velocity from the DEM validation data.

We next address the question of whether the "downward signal" observed during the warm season could be caused by north/south displacement, as suggested by the reviewer. To do this, we calculated the change in north/south velocity that would be required  to produce the observed magnitude of seasonal change in vertical velocity (-8.0 cm/yr). We first subtracted our January/February vertical velocity product (Figure 5 from the main text) from our September/October vertical velocity product to quantify the observed seasonal change in vertical velocity. We then projected this observed vertical seasonal change into the ascending LOS, and calculated the theoretical magnitude of north/south velocity change that would be required to achieve the same change in ascending LOS velocity.

The seasonal change in velocity over the moving area can either be explained by a -8.0 cm/yr change in the mean vertical velocity or a +62 cm/yr change in the mean north/south velocity (Figure R19). A much larger change in the north/south velocity would be needed because our LOS measurements are insensitive to north/south motion. Within the moving area, some pixels would require a northward change in velocity larger than 3 m/yr. The -8 cm/yr mean vertical change is far more likely, and can be explained by the well-documented physical process of seasonal ice melt in these areas (Table R1; Irvine-Fynn et al., 2011). The alternative explanation suggested by the reviewer would require a large seasonal transition to *rapid*, *upslope* motion over most of the moraine dam moving area, which has no physical basis and is not reflected in the DEM validation data.

[Figure]

**Figure R19. Change in vertical (second left) or north/south velocity (second right) required to produce the same seasonal change in LOS velocity (left). *v* is velocity, and the subscripts (*asc*, *ud*, and *ns*) correspond to the ascending LOS, up/down, and north/south direction, respectively. Note the expanded color range on the north/south velocity plot. The required change in velocity in the vertical direction has a much smaller magnitude than the required change in velocity in the vertical direction, as the LOS vector has a small northward component. The required change in velocity in the north/south direction would not be consistent with the average north/south velocity from the DEM validation dataset (right).**

In summary, we demonstrate that bias caused by the LOS decomposition has only a small effect on our results. We argue that the observed seasonal change in velocity is more likely caused by a small vertical acceleration due to downwasting than a large horizontal acceleration.

All of the above analysis is now included in the supplementary material.

In summary, while your study addresses an important topic, significant revisions and validations are required before it can be considered for publication. I hope these comments provide constructive guidance for improving your work.

We thank the reviewer for numerous insightful and constructive comments. We sincerely hope that the reviewer can appreciate the amount of effort required to prepare these detailed responses to mitigate their concerns, and the fact that the original methods, results, and conclusions remain largely unchanged. The revised text and supplementary material is greatly improved as a result of this effort, and we are confident that it will stand up to further scrutiny from both the InSAR and cryosphere communities.

Best regards

Jolivet, R., & Simons, M. (2018). A multipixel time series analysis method accounting for ground motion, atmospheric noise, and orbital errors. Geophysical Research Letters, 45(4), 1814-1824.

Casu, F., Manconi, A., Pepe, A., & Lanari, R. (2011). Deformation time-series generation in areas characterized by large displacement dynamics: The SAR amplitude pixel-offset SBAS technique. IEEE Transactions on Geoscience and Remote Sensing, 49(7), 2752-2763.

**References**

Allstadt, K. E., D. E. Shean, A. Campbell, M. Fahnestock, and S. D. Malone (2015), Observations of seasonal and diurnal glacier velocities at Mount Rainier, Washington, using terrestrial radar interferometry, *The Cryosphere*, *9*(6), 2219–2235, doi:10.5194/tc-9-2219-2015.

Bekaert, D. P. S., Walters, R. J., Wright, T. J., Hooper, A. J., & Parker, D. J. (2015). Statistical comparison of InSAR tropospheric correction techniques. *Remote Sensing of Environment, 170*, 40-47.

Beyer, R. A., Alexandrov, O., & McMichael, S. (2018). The Ames Stereo Pipeline: NASA's open source software for deriving and processing terrain data. *Earth and Space Science*, *5*(9), 537-548.

Bhushan, S. & Shean, D. (2025). uw-cryo/debris_cover_smb: Bugfix for disp_mgm_corr.py (0.4.1). Zenodo. https://doi.org/10.5281/zenodo.14816790

Bhushan, S., & Shean, D. (2021). Chamoli Disaster Pre-event 2-m DEM Composite: September 2015 (1.0) [Data set]. Zenodo. https://doi.org/10.5281/zenodo.4554647

Bhushan, S., Shean, D., Hu, J. Y. M., Guillet, G., & Rounce, D. R. (2024). Deriving seasonal and annual surface mass balance for debris-covered glaciers from flow-corrected satellite stereo DEM time series. *Journal of Glaciology*, *70*, e6.

Brencher, G., Handwerger, A. L., & Munroe, J. S. (2021). InSAR-based characterization of rock glacier movement in the Uinta Mountains, Utah, USA. *The Cryosphere, 15*(10), 4823-4844.

Brencher, G., Henderson, S. T., & Shean, D. E. (2025). Removing atmospheric noise from InSAR interferograms in mountainous regions with a convolutional neural network. *Computers & Geosciences, 194*, 105771.

Casu, F., Manconi, A., Pepe, A., & Lanari, R. (2011). Deformation time-series generation in areas characterized by large displacement dynamics: The SAR amplitude pixel-offset SBAS technique. *IEEE Transactions on Geoscience and Remote Sensing*, *49*(7), 2752-2763.

Cuffey, K.M., & Paterson, W.S.B. (2010). The Physics of Glaciers (Fourth Edition). Butterworth-Heinemann.

Driscoll, F.G. (1980). Wastage of the Klutlan Ice-Cored Moraines, Yukon Territory, Canada 1. *Quaternary Research, 14*(1), 31-49.

Etzelmüller, B. (2000). Quantification of thermo-erosion in pro-glacial areas-examples from Svalbard. *Zeitschrift für Geomorphologie*, 343-361.

Gray, L., Joughin, I., Tulaczyk, S., Spikes, V.B., Bindschadler, R., & Jezek, K. (2005). Evidence for subglacial water transport in the West Antarctic Ice Sheet through three-dimensional satellite radar interferometry. *Geophysical Research Letters*, *32*(3).

Gupta, V., Rakkasagi, S., Rajpoot, S., Imanni, H. S. E., & Singh, S. (2023). Spatiotemporal analysis of Imja Lake to estimate the downstream flood hazard using the SHIVEK approach. *Acta Geophysica*, *71*(5), 2233-2244.

Haritashya, U. K., Kargel, J. S., Shugar, D. H., Leonard, G. J., Strattman, K., Watson, C. S., ... & Regmi, D. (2018). Evolution and controls of large glacial lakes in the Nepal Himalaya. *Remote Sensing, 10*(5), 798.

Henderson, S. T., & Pritchard, M. E. (2013). Decadal volcanic deformation in the Central Andes Volcanic Zone revealed by InSAR time series. *Geochemistry, Geophysics, Geosystems, 14*(5), 1358-1374.

Henderson, S. T., Delgado, F., Elliott, J., Pritchard, M. E., & Lundgren, P. R. (2017). Decelerating uplift at Lazufre volcanic center, Central Andes, from AD 2010 to 2016, and implications for geodetic models. *Geosphere, 13*(5), 1489-1505.

Irvine-Fynn, T. D. L., Barrand, N. E., Porter, P. R., Hodson, A. J., & Murray, T. (2011). Recent High-Arctic glacial sediment redistribution: A process perspective using airborne lidar. *Geomorphology, 125*(1), 27-39.

Johnson, P.G. (1992). Stagnant glacier ice, St. Elias Mountains, Yukon. *Geografiska Annaler: Series A, Physical Geography, 74*(1), 13-19.

Joughin, I. (2002). Ice-sheet velocity mapping: a combined interferometric and speckle-tracking approach. *Annals of Glaciology*, *34*, 195-201.

Joughin, I., Smith, B., & Howat, I. (2018). A complete map of Greenland ice velocity derived from satellite data collected over 20 years. *Journal of Glaciology, 64*(243), 1-11.

Joughin, I., Smith, B. E., & Howat, I. (2018). Greenland Ice Mapping Project: ice flow velocity variation at sub-monthly to decadal timescales. *The Cryosphere*, *12*(7), 2211-2227.

Khadra, N. S. (2016, October 31). Nepal drains dangerous Everest lake. BBC News. https://www.bbc.com/news/world-asia37797559

King, L., & Volk, M. (1994). Glaziologie und Glazialmorphologie des Liefde-und Bockfjordgebietes, NW-Spitzbergen. *Zeitschrift für Geomorphologie.* (97), 145-159.

Kunz, J., Ullmann, T., & Kneisel, C. (2021). Internal structure and recent dynamics of a moraine complex in an alpine glacier forefield revealed by geophysical surveying and Sentinel-1 InSAR time series. *Geomorphology*, 108052.

Krüger, J., & Kjær, K.H. (2000). De-icing progression of ice-cored moraines in a humid, subpolar climate, Kötlujökull, Iceland. *The Holocene, 10*(6), 737-747.

Liu, H., Zhao, Z., & Jezek, K.C. (2007). Synergistic fusion of interferometric and speckle-tracking methods for deriving surface velocity from interferometric SAR data. *IEEE Geoscience and Remote Sensing Letters, 4*(1), 102-106.

Lukas, S., Nicholson, L. I., Ross, F. H., & Humlum, O. (2005). Formation, meltout processes and landscape alteration of high-Arctic ice-cored moraines—Examples from Nordenskiold Land, central Spitsbergen. *Polar Geography, 29*(3), 157-187.

Mattson, L.E. (1993). Ablation on debris covered glaciers: an example from the Rakhiot Glacier, Punjab, Himalaya. *Intern. Assoc. Hydrol. Sci., 218*, 289-296.

Mattson, L.E., & Gardner, J. S. (1991). Mass wasting on valley-side ice-cored moraines, Boundary Glacier, Alberta, Canada. *Geografiska Annaler: Series A, Physical Geography, 73*(3-4), 123-128.

Mckenzie, G.D. (1969). Observations on a collapsing kame terrace in Glacier Bay National Monument, south-eastern Alaska. *Journal of Glaciology, 8*(54), 413-425.

Notti, D., Davalillo, J. C., Herrera, G., & Mora, O. J. N. H. (2010). Assessment of the performance of X-band satellite radar data for landslide mapping and monitoring: Upper Tena Valley case study. *Natural Hazards and Earth System Sciences, 10*(9), 1865-1875.

Nuth, C., & Kääb, A. (2011). Co-registration and bias corrections of satellite elevation data sets for quantifying glacier thickness change. *The Cryosphere*, *5*(1), 271-290.

Østrem, G. (1965). Problems of dating ice-cored moraines. *Geografiska Annaler: Series A, Physical Geography, 47*(1), 1-38.

Richardson, S. D., & Reynolds, J. M. (2000). An overview of glacial hazards in the Himalayas. *Quaternary International*, *65*, 31-47.

Ross, A.B. (1976). A form and process response study of a terminal ice cored ablation moraine [Doctoral dissertation, University of Ottawa (Canada)].

Rouault, E., Warmerdam, F., Schwehr, K., Kiselev, A., Butler, H., Łoskot, M., Szekeres, T., Tourigny, E., Landa, M., Miara, I., Elliston, B., Chaitanya, K., Plesea, L., Morissette, D., Jolma, A., Dawson, N., Baston, D., de Stigter, C., & Miura, H. (2024). GDAL (v3.9.3). Zenodo. https://doi.org/10.5281/zenodo.13929593

Samsonov, S., Tiampo, K., & Cassotto, R. (2021). Measuring the state and temporal evolution of glaciers in Alaska and Yukon using synthetic-aperture-radar-derived (SAR-derived) 3D time series of glacier surface flow. *The Cryosphere*, *15*(9).

Sánchez-Gámez, P., & Navarro, F.J. (2017). Glacier surface velocity retrieval using D-InSAR and offset tracking techniques applied to ascending and descending passes of Sentinel-1 data for southern Ellesmere ice caps, Canadian Arctic. *Remote Sensing, 9*(5), 442.

Schmidt, D. A., & Bürgmann, R. (2003). Time-dependent land uplift and subsidence in the Santa Clara valley, California, from a large interferometric synthetic aperture radar data set. *Journal of Geophysical Research: Solid Earth*, *108*(B9).

Schomacker, A., & Kjær, K.H. (2008). Quantification of dead-ice melting in ice-cored moraines at the high-Arctic glacier Holmströmbreen, Svalbard. *Boreas, 37*(2), 211-225.

Shean, D. E., Alexandrov, O., Moratto, Z. M., Smith, B. E., Joughin, I. R., Porter, C., & Morin, P. (2016). An automated, open-source pipeline for mass production of digital elevation models (DEMs) from very-high-resolution commercial stereo satellite imagery. *ISPRS Journal of Photogrammetry and Remote Sensing*, *116*, 101-117.

Shean, D., Bhushan, S., Lilien, D., Knuth, F., Schwat, E., Meyer, J., Sharp, M., & Hu, M. (2023). dshean/demcoreg: v1.1.1 Compatibility and doc improvements (v1.1.1). Zenodo. https://doi.org/10.5281/zenodo.7730376

Somos-Valenzuela, M., McKinney, D. C., Byers, A. C., Voss, K., Moss, J., & McKinney, J. C. (2012). Ground penetrating radar survey for risk reduction at Imja Lake, Nepal. Center for Research in Water Resources, University of Texas at Austin. http://hdl.handle.net/2152/19751

Tonkin, T.N., Midgley, N.G., Cook, S.J., & Graham, D.J. (2016). Ice-cored moraine degradation mapped and quantified using an unmanned aerial vehicle: a case study from a polythermal glacier in Svalbard. *Geomorphology, 258*, 1-10.

UNDP. (2012). Community Based Flood and Glacial Lake Outburst Risk Reduction Project [Project Document]. https://www.undp.org/sites/g/files/zskgke326/files/migration/np/aa21b4ccde4230b2b26dda751438572149d185892d482a1c f417b3a1737ce05a.pdf

---

## Author Response (AR2)

Responses to reviewer and editor comments
**Quantifying degradation of the Imja Lake moraine dam with fused InSAR and SAR feature tracking time series**
George Brencher, Scott T. Henderson, David E. Shean
October 10, 2025

**Editor:**

Dear George Brencher and co-authors,

Thank you very much for your patience through this process. I well understand the challenges that come from these delays.

A second helpful review has now been received. While one review was positive and supported publication of the manuscript, the second pointed out that this is quite a long paper with two themes of method development and changes to the lake. A reviewer in the first round gave a similar assessment.

For the reasons that you expressed in your review response, I am sympathetic to your desire to keep this as a single manuscript. However, I can also see the benefit for the authors in splitting this manuscript into two and developing a more methods-based paper for a remote sensing journal. I will leave it to the authors to decide on this matter.

If you choose to continue with this single manuscript publication strategy, both reviews point to some recommendations about how to improve the manuscript. I agree with the second reviewer that more attention should be given to the method and less to the speculative aspect. This could be accomplished by shortening the discussion, for instance, by combining the "Ice melt" and "Ice flow" sections into a single section about the lake's seasonal dynamics. Additionally, I agree that the introduction could be streamlined and improved.

Should the authors move forward with this strategy, I will largely assess these writing and organisation matters in the next revision, as the results and scientific quality seem robust.

I wish the authors the best in moving forward with this.

Best regards,
Ian Delaney

Dear Editor,

Thank you for your persistence in finding an additional reviewer for this work. We would like to proceed with publication as a single manuscript. We appreciate the anonymous reviewer's critical reading of both our revised manuscript and previous reviews, and we have made additional revisions based on the insightful feedback we received. In particular, we shortened and streamlined the introduction and removed the more speculative parts of the discussion. The resulting manuscript is shorter and more focused on the methods. To allow for easy navigation of our discussion section, we maintained separate headings for the shortened "Ice melt" and "Ice flow" sections. On the whole, we feel that these revisions have made our manuscript more manageable for readers of *The Cryosphere*.

Sincerely,
George Brencher, on behalf of all authors

**Reviewer 3:**

I am writing this review taking both the revised manuscript and the reviews into account.
The paper presents different methods based on SAR data to investigate the stability and changes of the Imja Lake moraine dam. This is a highly relevant topic as continued degradation of ice-cored moraine lake dames increases the likelihood of a dam failure causing a glacial lake outburst flood. The work provides relevant insights into the degradation of the moraine, the methods applied are suitable and the results overall reliable so that the study should ultimately published.

It is also evident that the authors put major efforts in addressing both reviewers' concerns. This led to a clearly improved manuscript. In particular the inclusion of the comparison of the SAR-derived results and the intercomparison of the results of the different SAR methods are very valuable. Considering my knowledge (I am not an SAR expert but have some knowledge in SAR and InSAR and have processed SAR data), I'd judge the methods as sound, relevant influences affecting the accuracy (like the atmospheric noise, influence of the image acquisition geometry) considered and remaining uncertainties discussed. The comparison to the results of the high resolution imagery and results from previous published investigations show the overall reliability of the derived results. Moreover, the observed changes make sense also from the glaciological and geomorphological point of view. However, I am happy to leave the final judgement and possible suggestions for methodological improvements to a real SAR expert.

However, I have also some concerns regarding the current manuscript.

1. The manuscript combines the introduction a novel SAR based investigation, a detailed description of the relevance of the work and physical/glaciological interpretation of the observed results. These are different topics and from my point of view a bit too broad which leads to a quite lengthy manuscript and is also difficult to find the best journal for the current content. TC is a cryospheric journal, there is value to combine the introduction of a novel combination of the different SAR methods with a cryospheric application. However, as the focus is on introducing the remote sensing methods a Remote Sensing Journal focussing on relevant applications might be the better choice (The methodology could be well applied to other changes at the Earth surface not related to the cryosphere. I do not want to make an advertisement for specific journals, but there are several options of journals which are both read by remote sensing experts and cryospheric scientists. But certainly the study is also interesting for TC.

We appreciate the reviewer's thoughtful comments and opinions. We agree with many points, and touched on several aspects in our initial response to reviewers. Ultimately, we decided to proceed with TC, after streamlining the text.

2. The introduction is quite lengthy has some flaws (e.g. the use of the terminology, references etc.). I understand that one reviewer requested more details. However, from my point of few a one paragraph introducing the importance of investigating the dynamics/degradation of moraine dams will be enough (first introducing in few sentences the general importance of investigating GLOFs and then that weakening of the moraine dam is one of the major causes of GLOFs). In general, the introduction has some flaws as some to the scientific knowledge is not fully correctly summarised and some terms are not used correctly. In the first two sentences the authors first present the future of the glaciers and the past changes of the glacial lakes. It would make more sense to introduce both the current knowledge about the past glacier changes (e.g. as summarised in the recent GlaMBIE paper) and the glacial lakes (mention the cited reference, but also one recent for High Mountain Asia as Shugar et al. missed many glacial lakes). Then the potential future of the glaciers and glacial lakes can be mentioned.

We streamlined the introduction and removed the section on "moraine dam evolution." While we initially felt that this section contributed useful background information, we now agree that it contained more detail than necessary. In various other locations, we removed superfluous detail. The resulting introduction section is substantially shorter and more focused.

Reading the manuscript gives the impression that all moraines are ice-cored. This is not necessarily the case and also Shugar et al. (2020) do not mention ice-cored moraines. In addition, Ostrem, (1959) is not a suitable reference for the global occurrence of ice-cored moraines or moraines damming proglacial lakes which are addressed in this study. The term risk (L. 104) is incorrectly used. "Risk" in hazards is related to the potential for adverse impacts and includes hazard, exposure and vulnerability. The referencing is overall quite good. However, the referencing is a bit arbitrary, sometimes older relevant references are cited and the recent ones not or vice versa. There are few other issues which might not be present anymore when shortening the intro as suggested above. I would be happy to provide more a detailed review in this regard for a revised version.

We agree that readers could get the false impression from our introduction that all moraines contain buried ice. To address this issue, we removed the phrase "dammed by unstable ice-cored moraines" from the sentence in our first introductory paragraph (Line 29). We updated the following sentence in our second introductory paragraph to communicate that not all moraine dams contain buried ice:

"Where glacial lakes are dammed by moraines, hazard assessments frequently consider moraine dam stability, the presence of buried ice within moraine dams, potential GLOF triggering events, and downstream impacts (Rounce et al., 2016)."

We also removed the "Moraine dam evolution" section, which contained the Ostem (1959) reference and the incorrect use of the term "risk." We checked the full manuscript and found that this was the only instance of that word. This section also contained most of the older references, and removal improves the cohesiveness of citations.

Few suggestions for potentially relevant references which were not considered (for information only, you may decide to include or not):

Atwood et al. (2010). Using L-band SAR coherence to delineate glacier extent. Canadian Journal of Remote Sensing, 36(S1), S186-S195. https://doi.org/10.5589/m10-014

Frey et al. (2012). Compilation of a glacier inventory for the western Himalayas from satellite data: methods, challenges, and results. Remote Sensing of Environment, 124, 832–843. https://doi.org/10.1016/j.rse.2012.06.020

Huggel et al. (2002). Remote-sensing based assessment of hazards from glacier lake outbursts: a case study in the Swiss Alps. Canadian Geotechnical Journal, 39, 316–330.

Medeu et al. (2022). Moraine-dammed glacial lakes and threat of glacial debris flows in South-East Kazakhstan. Earth-Science Reviews, 229, 103999. https://doi.org/10.1016/j.earscirev.2022.103999 : The study includes figures which nicely show the existence of ice in moraine dams after outbursts.

Wangchuk et al. (2022). Monitoring glacial lake outburst flood susceptibility using Sentinel-1 SAR data, Google Earth Engine, and persistent scatterer interferometry. Remote Sensing of Environment, 271, 112910. https://doi.org/10.1016/j.rse.2022.112910

We updated the introduction to include these useful references:

Lines 43-46: Where moraine dam instability is not identified as a primary GLOF trigger mechanism, melting of buried ice can increase lake area, reduce width and height of dams, and provide potential pathways for seepage and piping (Richardson & Reynolds, 2000a; Emmer & Cochachin, 2013, Medeu et al., 2022).

Lines 50-53: Satellite remote sensing has been used to create glacial lake inventories, track glacial lake development (e.g. Fujita et al., 2009; Nie et al., 2018; Shugar et al., 2020), and recently, to monitor glacial lake dam and bank evolution (Haritashya et al., 2018; Scapozza et al., 2019; Wangchuk et al., 2022; Yang et al., 2022; Jiang et al., 2023; Yang et al., 2023; Yu et al., 2024).

Lines 76-79: InSAR coherence can also be used to identify significant change in surface characteristics, and low coherence has been used to map the extent of desert erosion (e.g. Cabré et al., 2020), landslides (e.g. Ohki et al., 2020; Jacquemart and Tiampo, 2021), flooding (e.g. Chini et al., 2019), and debris-covered glaciers (e.g. Atwood et al., 2010; Frey et al., 2012; Lippl et al., 2018).

L141ff: These two paragraphs do not really fit here. They contain mainly a description of the own methods and should be moved to and merged with the methods section.

We partially agree. The first paragraph largely reviews previous work on displacement time series processing and would be out of place in the methods section. The second paragraph largely relates to our method, and we moved the second paragraph to the methods section.

3. The discussion and interpretation contain some interesting and relevant aspects but is quite speculative. It makes sense to include some interpretation but they should not be too speculative and backed up by observations if possible. E.g. the authors write "Backwasting and thermokarst development should…" (L529). Evidence could be provided by the high-resolution imagery. Or "Other processes should … " (L549) "…may also be present…" L554. "…potentially experience…" L564. These are only few examples; there are several more.

We removed the more speculative sentences identified above and several others from the discussion, and streamlined interpretations that directly explain our observations.

In sum, I see the major strength in the article in the methodological part which can nicely applied in many aspects, the presented example being one important application. In this sense I have sympathy for the suggestion of splitting the content and submitting two to different journals. This would then also give the opportunity to be more specific on the methods without the manuscript being very long. And then improving the content related to GLOG hazards and degradation if an ice cored moraine.

Having written this I also see value in the combination as the authors argue and certainly leave it to the editor and authors to decide. Also for TC I suggest to shorten the manuscript and focus even more on the method and be less speculative.

We thank the reviewer for their helpful comments. For the reasons we outlined in response to Reviewer 1, we still prefer that this work remains a single manuscript, rather than being split up into two. After streamlining the introduction and removing the more speculative parts of the discussion, the manuscript is notably shorter and more focused on the methods and important cryospheric science results. We feel that these revisions have made the manuscript more manageable and interesting to readers of *The Cryosphere*.

---

## Author Response (AR3)

Responses to reviewer and editor comments
**Quantifying degradation of the Imja Lake moraine dam with fused InSAR and SAR feature tracking time series**

George Brencher, Scott T. Henderson, David E. Shean

November 29, 2025

**Editor:**

Dear George Brencher and co-authors,

Thank you for the work on this manuscript and for considering the reviewers' comments. Upon receiving the last comments, I am very happy to recommend the manuscript for publication. Congratulations!

The reviewer pointed out some matters in the reference list. Please address these, along with giving the manuscript a careful read.

Again, it is great to see this paper get published, and I certainly consider it a highly valuable contribution. Thank you for your efforts and patience through this process.

Best regards,
Ian

Dear Editor,

We're very happy to hear that the manuscript has been recommended for publication. We have addressed the issue with the references list. We have also done a careful read-through, and made minor edits to address grammar, typos, inconsistent terminology, and to make sure the manuscript complies with the style guide. We have also archived the code and data related to this work on Zenodo and added corresponding references. These edits have not changed the content of the manuscript. Thank you very much for all your efforts throughout this process!

Sincerely,

George Brencher, on behalf of all authors

**Reviewer 3:**

Dear authors,

thank you for cosidering my comments to further improve the interesting and relevant study.
If can be accepted now from my point of view. But please make sure that the reference list is updated before resubmission.

Best regards,

Tobias Bolch

Dear Reviewer,

Thank you again for your helpful comments. We have updated the reference list to include all the work cited in the text.

Best wishes,

George Brencher, on behalf of all authors